# Rectifying Diffusion Guidance with Exponential Moving Average

## Abstract

Continuous-time generative models, such as diffusion or flow models, employ Classifier-Free Guidance (CFG) to generate high-quality samples. However, CFG sacrifices sample diversity at the cost of improving sample quality, which leads to oversimplification. Furthermore, at high guidance scales, the excessive accumulation of guidance term causes oversaturation and leads to degraded sample quality. In this paper, we revisit the property that the Exponential Moving Average (EMA) of a model's predictions during sampling phase acts as a low-pass filter. By suppressing high frequencies, the EMA of a model's predictions inherently degrades sample quality without losing its low frequency information, which means still retaining conditional components, allowing it to be leveraged as the "weak" version. To exploit this property, we introduce Rectified EMA Guidance (REG), a simple yet effective training-free approach. REG rectifies guidance term with the difference between the EMAs of the model's conditional and unconditional predictions. This rectified guidance term offsets amplified conditional components, playing a crucial role in preserving sample diversity. We validated REG performance on both class-conditional and text-conditional models and demonstrated that it improves sample quality and preserves diversity by preventing oversaturation and oversimplification even in high-guidance scenarios. To the best of our knowledge, REG is the first training-free guidance method that improves sample quality orthogonally without requiring auxiliary components or specific model architecture, or modalities. Therefore, REG is applicable to a wide range of generative models, including large-scale public and industrial models.

## 1 Introduction

Generative models based on continuous-time dynamics, such as diffusion probabilistic models (DPMs) (Sohl-Dickstein et al. (2015); Song & Ermon (2019); Ho et al. (2020); Song et al. (2020); Karras et al. (2022)) and continuous normalizing flows (CNFs) (Chen et al. (2018); Lipman et al. (2022); Liu et al. (2022)), have shown remarkable success in high-fidelity data synthesis. DPMs generate data by reversing a stochastic denoising process using learned score functions, which are often formulated as Stochastic Differential Equations (SDEs) (Song et al. (2020)). On the other hand, CNFs use neural Ordinary Differential Equations (ODEs) (Chen et al. (2018)) to deterministically transport samples from a simple source distribution to the complex target distribution via learned velocity fields (Lipman et al. (2022); Liu et al. (2022)). Under the assumption of a Gaussian path, these two approaches can be unified within a probabilistic flow (PF) ODE framework (Song et al. (2020); Karras et al. (2022); Lipman et al. (2022); Esser et al. (2024)), providing a continuous-time interpretation of generative modeling for modeling the target distribution.

Sampling from the target distribution requires simulating the ODE which accumulates errors, due to the low-probability regions, the mean-predicting (marginalizing) nature of the model, and discretization errors, causing the sampled target distribution to deviate from the original target distribution. (Karras et al. (2024a)). To address this, guidance methods, such as Classifier Free Guidance (CFG) (Ho & Salimans (2021)), have been developed to heuristically nudge samples toward the guided direction by amplifying the guidance term. This term is constructed using the difference between conditional and unconditional predictions $u^\theta(x \mid y) - u^\theta(x)$. However, this guidance term includes both conditional and quality-improving components; consequently, neither can be improved orthogonally (Bradley & Nakkiran (2024); Zheng

& Lan (2024)). Therefore, as the guidance scale increases, both the conditional and quality improving components grow together, eventually leading to issues such as oversaturation and oversimplification (Saharia et al. (2022); Kynkäänniemi et al. (2024); Karras et al. (2024a)).

As it is well known in the naive truncation of GAN, overly amplified conditional signal improves conditional fidelity at the significant cost of the sample diversity (Marchesi (2017); Brock et al. (2019); Karras et al. (2024a)). Similar patterns emerge in CFG, which simultaneously boosts both conditional and quality improving components. Even though there are plenty of guidance methods, only few of them are formulated to alleviate this diversity preserving issue, particularly the overemphasis of the conditional direction. Interval Guidance (IG) (Kynkäänniemi et al. (2024)) restricts guidance to mid-sampling intervals, reducing monotonicity in earlier sampling steps. Adaptive Projected Guidance (APG) (Sadat et al. (2024b)) decomposes the guidance vector into semantic and orthogonal components, adjusting their contributions independently. Auto Guidance (AG) (Karras et al. (2024a)) takes a more explicit approach, isolating quality-relevant directions by comparing outputs from "strong" and "weak" conditional models. While effective, AG requires additional parameters for the "weak" conditional model, which should be carefully aligned with "strong" conditional model, limiting its scalability and compatibility with large scale industrial models where extra checkpoints are inaccessible.

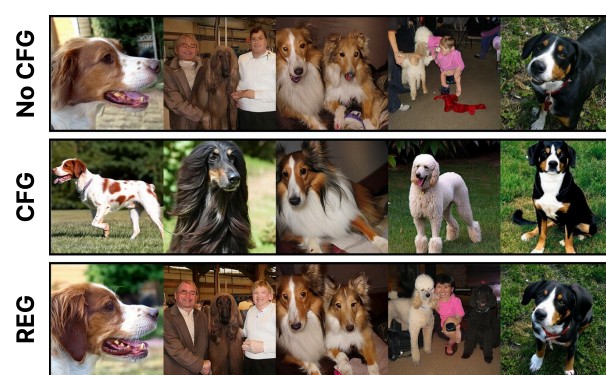

Figure 1: **Simplification in CFG.** In images generated from EDM2-xxl, CFG shows the side effect of oversimplifying the structure compared to an image without guidance (No CFG). On the other hand, our proposed REG successfully alleviates this phenomenon while enhancing the overall quality of the images.

To address this issue, we introduce Rectified EMA Guidance (REG), a training-free approach designed to improve image quality while preserving diversity, even at high guidance scales. REG achieves an effect comparable to that of AG, which utilizes a carefully aligned, "bad" version of itself to replace the unconditional term in CFG. However, unlike AG, REG does not require a "bad" version of the model, making it suitable for large-scale industrial models like SD3. We begin by analyzing the Exponential Moving Average (EMA) of the predicted velocity and demonstrate that its frequency distribution differs from that of the predicted velocity itself, leading to quality degradation. Since both guided and unguided EMA predictions result in reduced quality, we leverage the difference between guided EMA and unguided EMA as a guided component, with a diminished quality component. Finally, we rectify the unguided term of CFG by incorporating the new guided signal—the difference between guided EMA and unguided EMA, augmented with the unguided prediction. This process produces a weak guided term, which focuses on enhancing the quality component while suppressing the guide components of CFG, effectively preserving diversity while improving image quality and mitigating issues like oversaturation and oversimplification.

We validate REG across both class-guided (Karras et al. (2024b)) and text-guided (Podell et al. (2023); Esser et al. (2024)) models. For EDM2, we focused evaluating on generation quality and diversity, while for Stable Diffusion models, we additionally evaluate conditional fidelity (i.e., prompt alignment). Our experiments demonstrate that REG consistently outperforms or matches with the baseline methods across all metrics, even at high guidance scales where traditional CFG-based approaches struggle. Qualitatively, REG effectively mitigates oversaturation and oversimplification — the typical pitfalls of existing methods under strong guidance. It preserves the color richness and spatial layout seen at lower guidance scales, ensuring stable and faithful generation. Crucially, REG does not rely on additional, carefully aligned parameters like Auto Guidance (AG) does. This training-free nature makes REG directly applicable to publicly released models such as SDXL and SD3 where additional "weak" models are inaccessible.

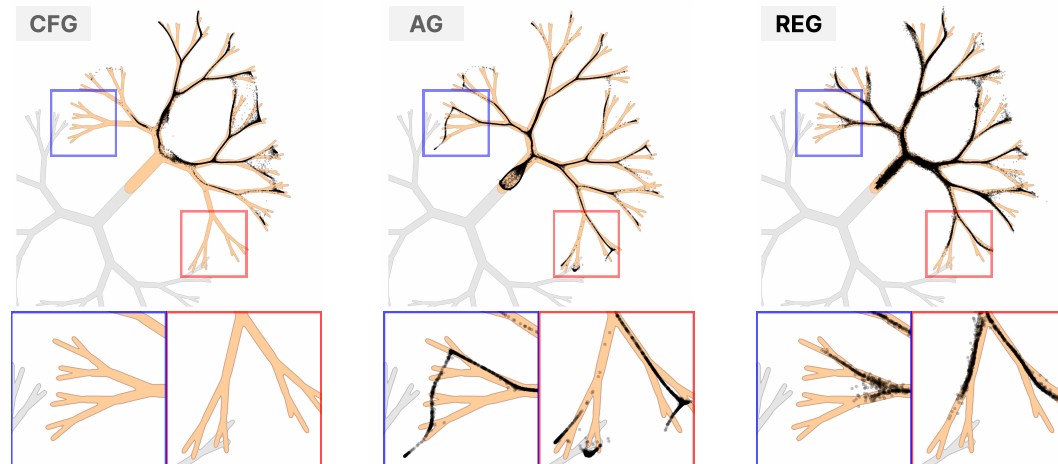

Figure 2: **Visualization of the Oversimplification Phenomenon in a 2D Toy Distribution.** Comparison of sampled 2d fractal-like distributions from (Karras et al. (2024a)) across different guidance methods. Orange denotes the target class for guidance, while gray denotes the non-target class. (Left) CFG removes outliers at the cost of diversity, losing the mode near the gray distribution. (Middle) AG preserves this mode, successfully maintaining diversity while removing outliers. (Right) Our method, REG, matches the performance of AG in preserving diversity without the need for an auxiliary guided weak model. Additional comparison between other guidance methods and detailed information about the 2D fractal-like distribution are provided in Sec.4 and Fig. 19 in Appendix.I

## 2 PRELIMINARIES

**Continuous-time Generative Models** Continuous-time generative models (Song et al. (2020); Lipman et al. (2022)) introduce sampling method for target sample by simulating an ODE that transports samples from a source distribution, $p_0(x_0)$, to a target distribution, $p_1(x_1)$, where $x_0, x_1 \in \mathbb{R}^d$ and $x_1 = z$. Two main categories are diffusion models (Ho et al. (2020); Karras et al. (2022)) which use a SDE or its deterministic counterpart, the probability flow ODE (PF-ODE), and flow models (Lipman et al. (2024); Liu et al. (2022)) which use an ODE based on a continuity equation.

In diffusion models, the score function $\nabla_x \log p_t(x|z)$ is approximated by a neural network, often using equivalent parameterizations like $\epsilon_t^\theta(x)$, $v_t^\theta(x|y)$, or $D_t^\theta(x|y)$ (Song & Ermon (2019); Ho et al. (2020)), where $y \in \mathbb{R}^m$ is a label or prompt for conditional generation. Similarly, flow models learn a conditional vector field $u_t^\theta(x|y)$ to transport samples. For a Gaussian probability path $p_t(x)$, the conditional probability path can be expressed as $p_t(x|z) = \mathcal{N}(x; \alpha_t z, \sigma_t^2 I)$ with a schedule $(\alpha_t, \sigma_t)$, which satisfies boundary conditions $\alpha_0 = \sigma_1 = 0$, $\alpha_1 = \sigma_0 = 1$, and the score and vector field become inter-convertible (Lipman et al. (2022); Esser et al. (2024)). This allows a unified ODE framework for modeling and sampling, where generation proceeds by solving the learned ODE backward in time from a Gaussian sample (Lu et al. (2022)).

However, directly using the conditional vector field $u_t^\theta(x|y)$ often results in degraded sample quality and conditional alignment, resulting in a sample far from the target distribution. To address this, CFG (Ho & Salimans (2021)) introduces a linear interpolation between unconditional and conditional fields:

$$\tilde{u}_t^\theta(x|y) = u_t^\theta(x) + (1 + w)(u_t^\theta(x|y) - u_t^\theta(x)) \tag{1}$$

where $w \in [0, \infty)$. While CFG and its variants (Kynkäänniemi et al. (2024); Karras et al. (2024a); Sadat et al. (2024b; 2025)) improve perceptual quality and semantic alignment, they often compromise sample diversity and introduce artifacts like oversaturation and oversimplification (Saharia et al. (2022); Karras et al. (2024a)).

### 2.1 EXPONENTIAL MOVING AVERAGE (EMA)

Exponential Moving Average (EMA) is a common technique for smoothing temporal signals by calculating exponentially weighted averages. For a discrete sequence $v_{t_k}$, EMA of model's prediction

$m_{t_k}$ is defined recursively as:

$$m_{t_k} = \beta m_{t_{k-1}} + \xi v_{t_k} = \beta^{k+1} m_0 + \xi \sum_{k=0}^{k} \beta^k v_{t_k} \tag{2}$$

where $\beta \in [0,1), \xi \in (0,1]$ are decay coefficients, $m_0$ is an initial momentum value that generally set to 0 and $k$ is the sequence index. This formula prioritizes recent values of $v_{t_k}$ while preserving a decaying memory of past values $v_{t_{i<k}}$. EMA can be extended to continuous time by interpreting it as a first-order low-pass filter. The corresponding differential equation is given by $\tau \frac{dm(t)}{dt} = -m(t) + v(t)$, where $\tau > 0$ is the time constant that controls the smoothness. The solution to this ODE is $m(t) = \frac{1}{\tau} \int_0^t e^{-(t-s)/\tau} v(s) \, ds$, which expresses the EMA as a convolution of $v(t)$ with an exponential kernel. This continuous-time version allows EMA to be easily used into continuous-time generative models such as those based on ODEs.

EMA has been adopted in diffusion models in some works to stabilize score or velocity predictions across time (Dockhorn et al. (2021); Wizadwongsa et al. (2023); Qian et al. (2024); Sadat et al. (2024b); Ma et al. (2025)). In this work, we focus on the role of EMA in sampling phase, where it acts as a temporal filter to suppress high frequencies, while preserving the low frequencies.

## 3 METHOD

Before discussing *Sampling with EMA*, we first address the limitations of CFG at high guidance scales. Beyond reducing image diversity as shown in a recent study(Karras et al. (2024a)), high guidance can also introduce two artifacts, oversaturation and oversimplification. Oversaturation arises from excessive amplification of the conditional signal. This leads to unnatural visuals, where colors appear exaggerated and details may become obscured, ultimately lowering overall image quality. Oversimplification refers to global structural distortions caused by the amplified signal shifting samples within the semantic manifold. While CFG enhances fidelity, it may compromise visual balance.

This analysis of CFG provides a foundation for our proposed REG. The method section first introduces EMA as a temporal low-pass filter. The low-frequency components of the model's predictions exhibit stability(Sadat et al. (2025)) and hold properties related to diversity. While effective at noise mitigation, EMA can suppress fine-scale details. This understanding of EMA's properties is then leveraged and corrected in the REG framework to reduce oversaturation and preserve layout while enhancing image fidelity.

### 3.1 SAMPLING WITH EMA

From a signal processing standpoint, EMA can be modeled as a first-order infinite impulse response low-pass filter (Li et al. (2024)), attenuating rapid fluctuations in the predicted velocity field. Its recursive formulation is given by $m_t = \beta m_{t-1} + v_t$, where $\beta \in (0,1)$ governs the exponential decay of past samples. This corresponds to the $z$-domain transfer function: $H(z) = \frac{1}{1 - \beta z^{-1}}$. As $\beta$ approaches 1, the filter integrates a longer temporal history, damping fluctuations across the entire spectrum. Evaluating this transfer function on the unit circle $z = e^{j\omega}$ (Oppenheim & Willsky (1997)) yields the magnitude response:

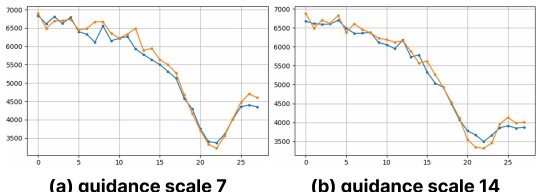

(a) guidance scale 7     (b) guidance scale 14

Figure 3: **Frequency energy comparison**. Graph comparing the evolution of high-frequency energy, which is critical for image details, during the sampling process of CFG (blue) and REG (orange). REG exhibits a larger uptrend in high-frequency energy in the final steps, indicating enhanced generation of fine details compared to CFG. The energy was calculated as the L2 norm of the high-frequency components from the FFT.

$$|H(e^{j\omega})| = \frac{1}{\sqrt{1 - 2\beta \cos\omega + \beta^2}}, \quad |H(e^{j\pi})| \ll |H(e^{j0})|. \tag{3}$$

This response confirms that high-frequency components ($\omega \approx \pi$) are attenuated relative to low-frequency content ($\omega \approx 0$), with stronger suppression for larger $\beta$. However, this temporal filtering

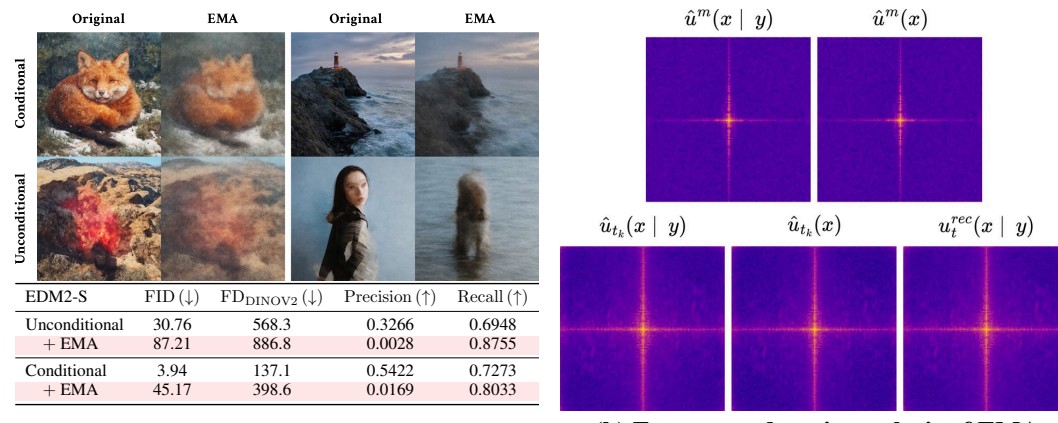

(a) EMA effect on perceptual quality     (b) Frequency domain analysis of EMA

Figure 4: **LPF property of EMA. (Left)** The images sampled with Algorithm 3. shows degraded sample quality both perceptually and quantitatively. While, the sampled images keep coarse layout but suppress sharpness and fine detail, indicating the preservation of low frequencies; the table below also shows higher FID/FD and lower Precision, which means degraded sample quality. **(Right)** Frequency characteristics of the EMA-rectified velocity align with vanilla predictions, whereas EMA-applied velocities differ. Specifically, the top row shows EMA-applied velocities $\hat{u}^m(x \mid y)$ and $\hat{u}^m(x)$; the bottom row shows vanilla/original $\hat{u}_{t_k}(x \mid y)$, $\hat{u}_{t_k}(x)$, and the EMA-rectified prediction $u_t^{rec}(x \mid y)$. Per-timestep frequency characteristics of each component are shown in Fig. 20.

propagates to the spatial domain during reverse diffusion, often diminishing intricate details. This interpretation uses a simplified non-linear model like U-Net, which is detailed in the Appendix.

Our empirical analysis, detailed in the Appendix, demonstrates that applying an EMA to velocity predictions results in excessive output smoothing, thereby compromising the structural and semantic fidelity of the generated images. As illustrated in Fig. 4 (a), this smoothing effect attenuates temporal fluctuations. Also, detailed sampling with EMA algorithm is elaborated in Algorithm 3.

## 3.2 Rectified EMA Guidance (REG)

CFG employs the difference, $\hat{u}_{t_k}(x \mid y) - \hat{u}_{t_k}(x)$, as the guiding signal. This velocity inherently captures not only the semantic fidelity associated with the target label $y$, but also reflects discrepancies in perceptual quality (Ho & Salimans (2021); Saharia et al. (2022); Karras et al. (2024a)). Such discrepancies arise due to the broader and more complex nature of the unconditional modeling task compared to the conditional counterpart (Karras et al. (2024a)). As the guidance scale $w$ increases, both semantic fidelity and perceptual quality are improved; however, beyond a certain threshold, semantic components are excessively amplified, leading to degradation phenomena such as oversaturation and oversimplification (Saharia et al. (2022); Marchesi (2017); Brock et al. (2019)).

By the definition of discretized EMA, we define the EMA predictions at $t_k$ timestep as below:

$$\hat{u}^m(x \mid y) := \sum_{s=0}^{k} \beta^s \hat{u}_{t_{k-s}}(x \mid y) \quad (4) \qquad \hat{u}^m(x) := \sum_{s=0}^{k} \beta^s \hat{u}_{t_{k-s}}(x) \quad (5)$$

Both $\hat{u}^m(x \mid y)$ and $\hat{u}^m(x)$ are obtained by temporally averaging the model predictions $\hat{u}_{t_k}(x \mid y)$ and $\hat{u}_{t_k}(x)$, which suppresses high-frequency signals and typically results in blurred outputs with degraded perceptual quality and contains the low-frequency component of the conditional prediction. This counteracts the CFG guidance inducing an opposing shift on the image manifold. Constructing a rectified term from this component frames our approach as a form of weak model guidance. When the EMA-based conditional and unconditional predictions become too similar, the resulting outputs exhibit nearly identical spectral distributions (Fig. 4.(b), top row), thereby diminishing the overall perceptual fidelity of both predictions. Empirical results further confirm that the perceptual discrepancy between them is reduced compared to their non-EMA counterparts. The difference $\hat{u}^m(x \mid y) - \hat{u}^m(x)$ predominantly captures the semantic shift, with perceptual quality components being relatively suppressed, resulting in a directionally consistent semantic signal.

Based on this observation, we define a rectified prediction:

$$u_t^{rec}(x \mid y) = \hat{u}_t(x) + \gamma(s_{t_k}^c \hat{u}^m(x \mid y) - s_{t_k}^u \hat{u}^m(x)), \qquad (6)$$

where $\gamma \in [0.1, 1.0]$ is a rectifying scale, $s_{t_k}^c = \frac{\|\hat{u}_{t_k}(x|y)\|}{\|\hat{u}^m(x|y)\|}$ and $s_{t_k}^u = \frac{\|\hat{u}_{t_k}(x)\|}{\|\hat{u}^m(x)\|}$ are scaling factors for $\hat{u}^m(x|y)$ and $\hat{u}^m(x)$ respectively. These scaling factors are introduced to match the magnitude of the EMA-based predictions to that of the original predictions, ensuring that the EMA-derived semantic shift operates on a comparable scale. By incorporating the scaled EMA-induced semantic shift, $s_{t_k}^c \hat{u}^m(x \mid y) - s_{t_k}^u \hat{u}^m(x)$ into the unconditional prediction $\hat{u}_t(x)$, the resulting $\tilde{u}_t(x)$ maintains a spectral distribution similar to $u_t(x)$ (Fig 4.(b) bottom row), while enhancing semantic fidelity and mitigating perceptual artifacts. Notably, this approach requires no additional training or auxiliary model, in contrast to methods such as Auto Guidance (AG) (Karras et al. (2024a)), which relies on weak models often unavailable in industrial-scale models like SD3 (Esser et al. (2024)). Empirical results demonstrate that the proposed method preserves perceptual quality while maintaining a level of sample diversity comparable to AG.

Finally, we amplify the rectified prediction $u_t^{rec}(x \mid y)$ as a guiding signal, which is equal to replacing the original CFG guidance term $\hat{u}_{t_k}(x \mid y) - \hat{u}_t(x)$ with $\hat{u}_{t_k}(x \mid y) - u_t^{rec}(x \mid y)$ as below:

$$\tilde{u}_t(x \mid y) = u_t^{rec}(x \mid y) + w(\hat{u}_{t_k}(x \mid y) - u_t^{rec}(x \mid y)) \qquad (7)$$

As a result, the REG guidance term offsets excessive semantic components during prediction and more effectively balances perceptual quality and diversity compared to standard CFG. Given that APG Sadat et al. (2024b) shares the usage of EMA, we discuss their theoretical differences in Appendix G. In particular, at high guidance scales where semantic components becomes overly dominant, REG alleviates degradation phenomena such as oversimplification and oversaturation (Fig. 3). The entire REG algorithm is detailed in Algorithm 1.

## 4 EXPERIMENTS

We evaluate REG on ImageNet 512 (with EDM2-s and -xxl) and MS-COCO 5K (with Stable Diffusion-XL and -3), assessing image quality (**FID**, **FD$_{\text{DINOv2}}$**, **Precision**), sample diversity (**Recall**), and text-image alignment (**CLIPScore**). For the EDM2, we used the checkpoint from AG's best settings, prioritizing a direct comparison with AG over using an optimally aligned conditional/unconditional pair. Also, to visualize the sampling distributions, we leverage the 2D fractal-like Gaussian mixture toy example from (Karras et al. (2024a)), which reflects the properties of actual image manifold (low local dimensionality and hierarchical emergence of local detail). Additional experimental setups are detailed in Appendix. First, in 4.1 we conduct an experiment that the proposed rectified term contains conditional components, allowing the strength of the conditional component of REG can be controlled by $\gamma$ scale. Second, in 4.2, we used the proposed rectified term to offset the overly accumulated conditional component of CFG on various guidance scales $w$, to improve sample quality and preserve sample diversity. Finally, in 4.3, we compare REG with other guidance methods which are formulated to improve perceptual quality.

### 4.1 CONTROLLABLE CONDITIONAL COMPONENT

In this section, Fig. 5, we conducted experiments to validate our central hypothesis: the proposed rectified term, $u_t^{rec}(x \mid y)$ (Eq. 6), contains a controllable conditional component. To test this, we designed an analysis centered on a scaling parameter, $\gamma$, which modulates the influence of this term. We investigate whether varying $\gamma$ can offset the conditional component in two distinct domains: high-dimensional text-to-image generation and a 2D Gaussian mixture distribution from Karras et al. (2024a).

**Analysis on Text-to-Image Generation**   As detailed in Fig. 5, 6, we first evaluate the effect of $\gamma$ in text-to-image synthesis. Qualitatively, we observe that increasing $\gamma$ effectively mitigates common artifacts of excessive conditioning, such as oversaturation and oversimplification in the generated images. This provides visual evidence that our rectified term counteracts the negative side effects of strong guidance. This visual finding is supported by our quantitative analysis, where we assess sample diversity using Recall. The results show that the drop in Recall for REG is significantly smaller than for standard CFG as the guidance scale increases, confirming that our method preserves diversity by successfully offsetting the harmful aspects of the conditional component.

| 0.2 | 0.6 | 1.0 | 1.4 | 1.8 | Unconditional |

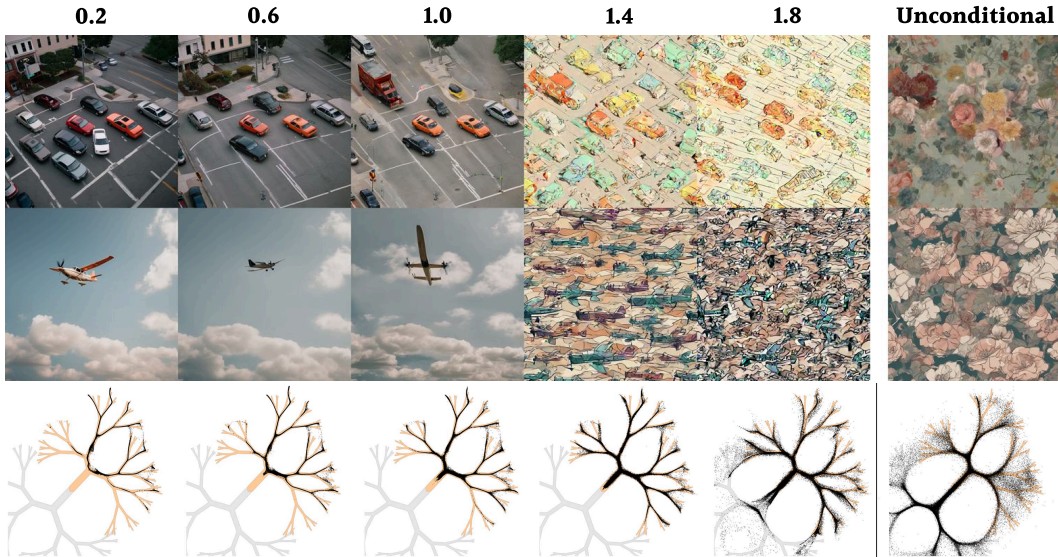

Figure 5: **The Offsetting Effect of** $\gamma$ **on the Conditional Component** We demonstrate that the conditional component can be controlled by varying $\gamma$. Qualitatively, visual examples with a fractal-like toy example illustrate that a sufficiently large $\gamma$ successfully cancels the conditional component, leading to a sample distribution nearly identical to the unconditional case, but with enhanced quality (sharper) (*1.8 vs unconditional*).

**Analysis on 2D Gaussian Mixture Distribution** To visualize the underlying distributional effects, we conducted an experiment on a 2D Gaussian mixture distribution shown in the bottom row of Fig. 5. As $\gamma$ increases from zero, REG begins to cover branches of the distribution that were previously dropped (orange), demonstrating a clear improvement in sample diversity and

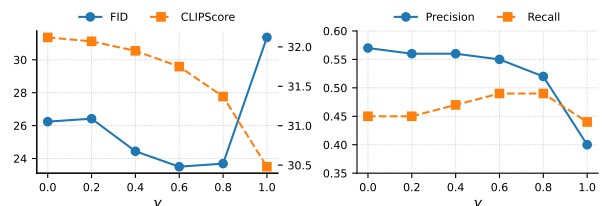

Figure 6: Effect of $\gamma$ on SDXL: CLIPScore declines as $\gamma$ increases, while FID remains stable up to a certain point.

mode coverage. As $\gamma$ is further increased by a large margin, the sampled distribution converges towards the unconditional target distribution (gray).

## 4.2 DIVERSITY PRESERVING PROPERTY OF REG

This section evaluates REG's primary advantage: its ability to improve sample quality while mitigating the loss of sample diversity—a critical trade-off in guided generation. To validate this, we present both qualitative and quantitative comparisons in Fig. 7. Our twofold analysis involves using a 2D Gaussian mixture distribution to visualize effects on the entire sample space and a text-to-image generation task to examine changes in individual samples.

**Analysis on 2D Gaussian Mixture Distribution** We analyze the overall distributional effects using a 2D Gaussian mixture. For standard CFG (third row of Fig. 7), increasing the guidance scale ($w$) enhances sample quality but leads to mode collapse, where an excessive accumulation of the conditional component causes entire branches of the target distribution to be dropped. In contrast, REG (fourth row) successfully mitigates branch dropping, thereby preserving the diversity of the target distribution. This visual observation is strongly supported by our quantitative results on the MS-COCO dataset. As the guidance scale and sample quality increase, the Recall metric—a proxy for sample diversity—decreases sharply for CFG. For REG, however, the decrease in Recall is significantly more gradual. This confirms that REG effectively navigates the quality-diversity trade-off, achieving high-fidelity generation without a substantial sacrifice in sample variety.

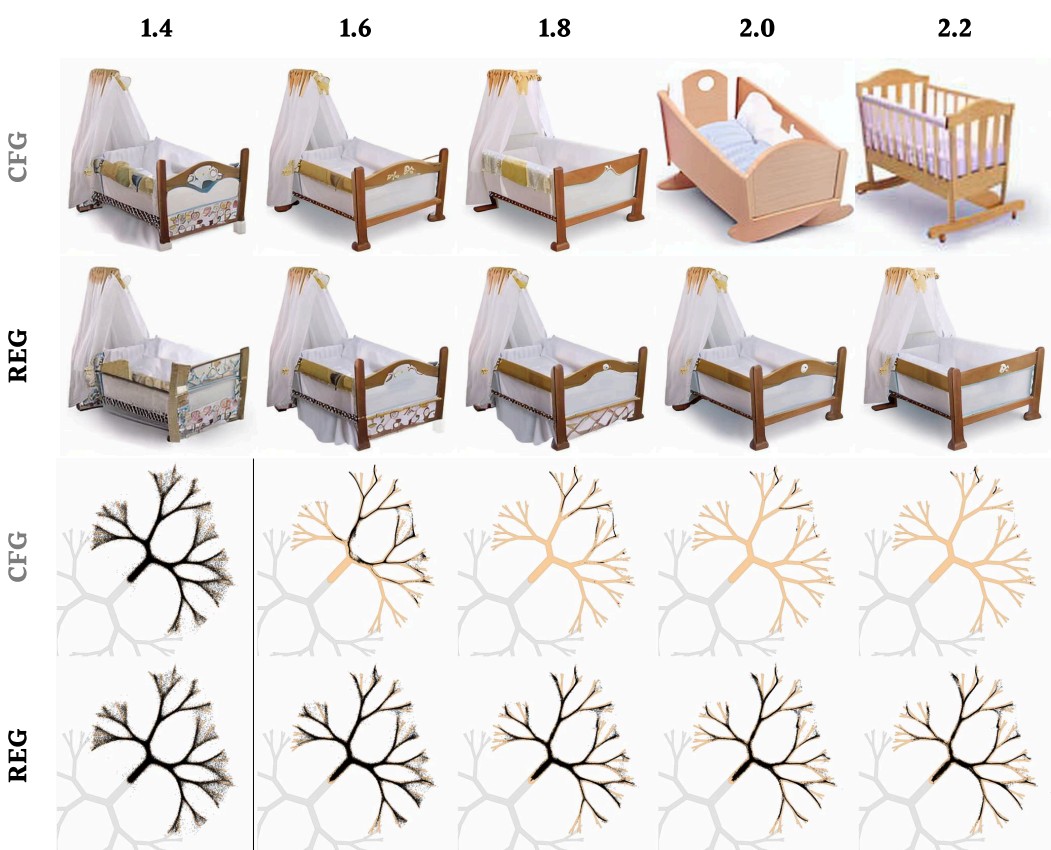

Figure 7: **Qualitative Diversity-Preserving Behavior of REG on EDM2-xxl.** As the guidance scale $w$ increases, standard CFG progressively collapses the underlying structure, dropping branches and oversimplifying the sample space (**first and third rows**). In contrast, REG preserves the global structure and maintains the coverage of the multi-modal distribution (**second and fourth rows**), demonstrating its ability to retain diversity even under strong guidance.

**Analysis on Text-to-Image Generation** We also visualize this trade-off on text-to-image task. With CFG (first row), increasing the guidance scale leads to oversimplification, where the complex structure of an image generated at a low scale ($w = 1$, far left) degrades into a simplistic form at a high scale (far right). REG (second row), however, enhances image quality without significantly distorting the image's core structure, preserving the details present at lower guidance scales. Fig. 8 demonstrate the quality-diversity trade-off. CFG sacrifices sample diversity for quality at high guidance scales (a rapid decrease in Recall), whereas REG preserves diversity more effectively (a slight decrease in Recall). Additional results are in Fig. 11, 16, 17, 31 and Table 3, 4

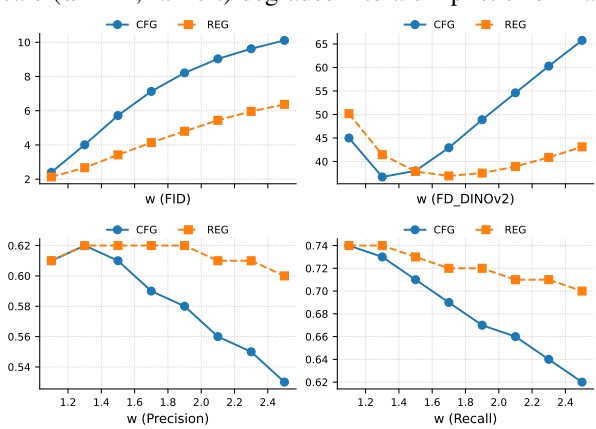

Figure 8: Quantitative Diversity-Preserving Behavior of REG on EDM2-xxl.

### 4.3 PERCEPTUAL COMPARISON WITH OTHER GUIDANCE METHODS

Finally, we compare REG against other quality-focused guidance methods (CFG Ho & Salimans (2021), APG Sadat et al. (2024b), FDG Sadat et al. (2025), IG Kynkäänniemi et al. (2024), TCFG Kwon et al. (2025), CFG++ Chung et al. (2024), and AG Karras et al. (2024a)) in Fig. 9

| REG | CFG | APG | FDG | IG | TCFG | CFG++ |
|-----|-----|-----|-----|-----|------|-------|

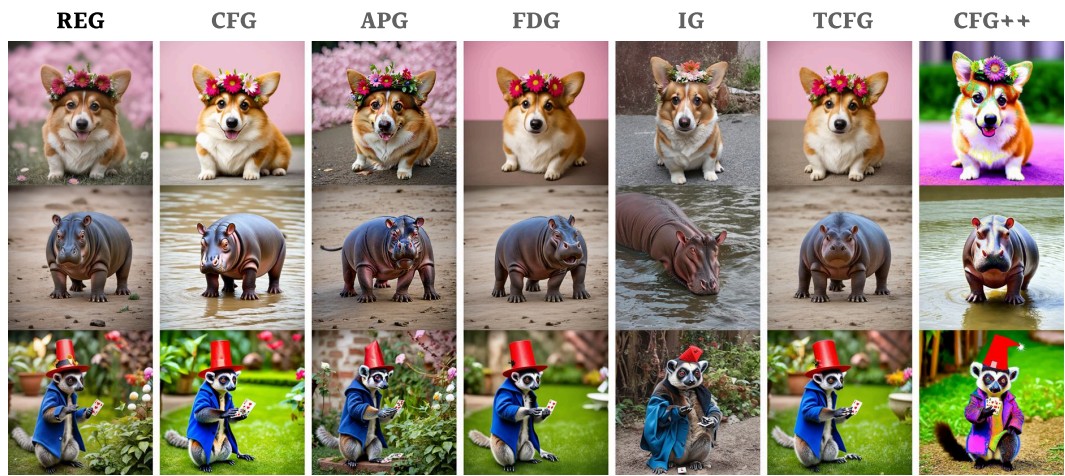

Figure 9: **Qualitative comparison of guidance methods on SD3 models.**

| EDM2-S | Conditional | REG | CFG | APG | FDG | IG | TCFG | CFG++ | AG |
|--------|-------------|-----|-----|-----|-----|-----|------|-------|-----|
| **FID** ($\downarrow$) | 3.93 | 2.04 | 2.07 | 3.05 | 3.94 | 2.09 | 2.12 | 2.87 | 1.38 |
| **FD$_{\text{DINOv2}}$** ($\downarrow$) | 137.12 | 69.16 | 72.10 | 52.90 | 137.13 | 59.94 | 72.37 | 112.17 | 50.63 |
| **Precision** ($\uparrow$) | 0.54 | 0.62 | 0.62 | 0.63 | 0.54 | 0.62 | 0.61 | 0.56 | 0.61 |
| **Recall** ($\uparrow$) | 0.73 | 0.72 | 0.71 | 0.69 | 0.73 | 0.71 | 0.71 | 0.74 | 0.71 |

| SD3 | Conditional | REG | CFG | APG | FDG | IG | TCFG | CFG++ | AG |
|-----|-------------|-----|-----|-----|-----|-----|------|-------|-----|
| **FID** ($\downarrow$) | 46.87 | 27.66 | 28.50 | 29.14 | 31.08 | 33.10 | 29.91 | 32.20 | - |
| **CLIPScore** ($\uparrow$) | 29.49 | 31.43 | 31.51 | 31.60 | 31.66 | 30.58 | 31.56 | 31.36 | - |
| **Precision** ($\uparrow$) | 0.12 | 0.57 | 0.55 | 0.54 | 0.51 | 0.19 | 0.58 | 0.50 | - |
| **Recall** ($\uparrow$) | 0.39 | 0.34 | 0.25 | 0.37 | 0.40 | 0.45 | 0.39 | 0.20 | - |

Table 1: **Comparison of guidance methods on EDM2-s and SD3 models.** For EDM2-s, each method is equipped with a class-conditional EDM2-s model and evaluated on ImageNet 512. For SD3, each method is evaluated on an SD3 model using a MS-COCO 5K subset. Note that AG is not available for SD3.

and Table 1. We group baselines into three categories: (1) methods that consider diversity preservation, (2) those that do not, and (3) AG, a special case requiring an aligned weak model, which makes it compatible with EDM2 but not SD3. In the table, we mark each method to indicate the public availability of its official code and hyperparameters; for methods with unreleased model-specific configurations, we report the best results from a local grid search. Further experimental details are provided in the Appendix. Both qualitative and quantitative results demonstrate REG's strong performance. Visually, images generated by REG match or surpass the quality of competing methods while maintaining superior sample diversity. FID, FD$_{\text{DINOv2}}$, and Precision, demonstrated REG's ability to generate high-quality images. More additional results are in Fig. 13, Fig. 14, Fig. 15 and Table 5.

## 5 CONCLUSION

In this work, we revisit the low-pass filter (LPF) property of the Exponential Moving Average (EMA) during the sampling phase of continuous-time generative models. Building on this analysis, we propose Rectified EMA Guidance (REG), a novel method that leverages the difference between EMA predictions to construct a rectified guidance signal. This signal successfully offsets the excessive conditional components inherent in Classifier-Free Guidance, enabling REG to enhance sample quality while preserving sample diversity. Crucially, REG is a training-free approach that requires neither external "weak" models nor model architectural assumptions, making it a practical and scalable solution that is readily applicable to large-scale industrial models.

## REPRODUCIBILITY STATEMENT

Our work is grounded in reproducible source code. We utilize publicly available and officially released implementations of the pretrained models mentioned in this paper. To facilitate the precise replication of our proposed method, we have included a detailed description of our REG implementation in Algorithm 1 and Appendix C. A comprehensive list of all hyperparameters and additional implementation details can be found in the Appendix B. To further enhance reproducibility, the source code used for our experiments is provided as part of the supplementary material.

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

## A    ALGORITHM

From a signal processing perspective, applying EMA to predicted velocity fields during sampling can be interpreted as imposing a first-order low-pass filter, attenuating high-frequency components while preserving lower frequencies. However, it introduces a frequency scale mismatch between the raw predictions and their EMA counterparts. As shown in Eq. (2), the magnitude response of the EMA filter diminishes sharply at higher frequencies, leading to the suppression of fine-scale structures such as edges and textures. Consequently, when EMA predictions are used to guide sampling, the resulting trajectory is biased toward oversmoothed solutions, often failing to capture the intricate, high-frequency details present in the original prediction. This frequency misalignment between the unfiltered and EMA-filtered signals undermines the effectiveness of EMA as a sampling guide, particularly in tasks requiring high structural or semantic fidelity. The red highlighted part of *CFG* algorithm 2 is modified like the blue highlighted part of *Sampling with EMA* algorithm 3 and *REG* algorithm 1.

---

**Algorithm 1 Rectified EMA Guidance (REG)**

---

**Input**: $x_0 \sim p_0, \{t_0 = 0, \ldots, t_N = 1\}, \beta = 0.9$
**Output**: $x_1 \sim p_1$

1: Let $x \leftarrow x_0, \hat{u}^m(x \mid y) \leftarrow 0, \hat{u}^m(x) \leftarrow 0$.
2: **for** $t_k = t_0$ to $t_N$ **do**
3:     $\hat{u}_{t_k}(x \mid y) \leftarrow u_{t_k}^\theta(x \mid y)$
4:     $\hat{u}_{t_k}(x) \leftarrow u_{t_k}^\theta(x)$
5:     $\hat{u}^m(x \mid y) \leftarrow \beta \hat{u}^m(x \mid y) + \hat{u}_{t_k}(x \mid y)$
6:     $\hat{u}^m(x) \leftarrow \beta \hat{u}^m(x) + \hat{u}_{t_k}(x)$
7:     $s_{t_k}^c \leftarrow \frac{\|\hat{u}_{t_k}(x|y)\|}{\|\hat{u}^m(x|y)\|}$
8:     $s_{t_k}^u \leftarrow \frac{\|\hat{u}_{t_k}(x)\|}{\|\hat{u}^m(x)\|}$
9:     $u_t^{rec}(x \mid y) \leftarrow \hat{u}_{t_k}(x) + \gamma(s_{t_k}^c \hat{u}^m(x \mid y) - s_{t_k}^u \hat{u}^m(x))$
10:     $\tilde{u}_{t_k}(x \mid y) \leftarrow u_t^{rec}(x \mid y) + (1+w)(\hat{u}_{t_k}(x \mid y) - u_t^{rec}(x \mid y))$
11:     $x \leftarrow x + (t_{k+1} - t_k)\tilde{u}_{t_k}(x \mid y)$
12: **end for**
13: **return** $x$

---

**Algorithm 2 CFG**

---

**Input**: $x_0 \sim p_0, \{t_0 = 0, \ldots, t_N = 1\}$
**Output**: $x_1 \sim p_1$

1: Let $x \leftarrow x_0$.
2: **for** $t_k = t_0$ to $t_N$ **do**
3:     $\hat{u}_{t_k}(x \mid y) \leftarrow u_{t_k}^\theta(x \mid y)$
4:     $\hat{u}_{t_k}(x) \leftarrow u_{t_k}^\theta(x)$
5:     $\tilde{u}_{t_k}(x \mid y) \leftarrow \hat{u}_{t_k}(x \mid y) + w(\hat{u}_{t_k}(x \mid y) - \hat{u}_{t_k}(x))$
6:     $x \leftarrow x + (t_{k+1} - t_k)\tilde{u}_{t_k}(x \mid y)$
7: **end for**
8: **return** $x$

---

**Algorithm 3 Sampling with EMA**

---

**Input**: $x_0 \sim p_0, \{t_0 = 0, \ldots, t_N = 1\}, \beta$
**Output**: $x_1 \sim p_1$

1: Let $x \leftarrow x_0, \hat{u}^m(x \mid y) \leftarrow 0$.
2: **for** $t_k = t_0$ to $t_N$ **do**
3:     $\hat{u}_{t_k}(x \mid y) \leftarrow u_{t_k}^\theta(x \mid y)$
4:     $\hat{u}^m(x \mid y) \leftarrow \beta \hat{u}^m(x \mid y) + \hat{u}_{t_k}(x \mid y)$
5:     $s \leftarrow \frac{\|\hat{u}_{t_k}(x|y)\|}{\|\hat{u}^m(x|y)\|}$
6:     $\tilde{u}_{t_k}(x \mid y) \leftarrow \hat{u}_{t_k}(x \mid y) + w(\hat{u}_{t_k}(x \mid y) - s\hat{u}^m(x \mid y))$
7:     $x \leftarrow x + (t_{k+1} - t_k)\tilde{u}_{t_k}(x \mid y)$
8: **end for**
9: **return** $x$

## B  ADDITIONAL EXPERIMENTAL DETAILS

We provide detailed descriptions of our experimental setup and implementation specifics for reproducibility.

**Experimental setups.**   To validate the effectiveness of REG, we compare it with various training-free guidance methods, including CFG Ho & Salimans (2021), Adaptive Projective Guidance (APG) Sadat et al. (2024b), Frequency-Decoupled Guidance (FDG) Sadat et al. (2025), Auto Guidance (AG) Karras et al. (2024a), Interval Guidance (IG) Kynkäänniemi et al. (2024), Tangential CFG (TCFG) Kwon et al. (2025), and CFG++ Chung et al. (2024). In addition, attention-based guidance methods such as Perturbed-Attention Guidance (PAG) Ahn et al. (2024), and Smooth Energy Guidance (SEG) Hong (2024) are compared on SDXL Podell et al. (2023). In Table 2, official hyperparameter settings from original papers or code are highlighted in ***bold***, and official baselines are underlined. Otherwise, a mild grid search was conducted to select the setting with the highest metric. We evaluate EDM2 on the ImageNet 512 dataset, a class-conditional image generation benchmark used for assessing sample fidelity. On the other hand, for Stable Diffusion, we adopt the MS-COCO 5K subset, which consists of image-caption pairs and is commonly used to evaluate text-to-image alignment performance. We evaluate REG from three perspectives. Image quality is measured using Fréchet Inception Distance (**FID**) Heusel et al. (2017), **FD$_{\text{DINOv2}}$** Stein et al. (2023) and **Precision** Kynkäänniemi et al. (2019). Sample diversity is assessed with **Recall** Kynkäänniemi et al. (2019). Conditional fidelity (Text-image alignment) is evaluated using **CLIPScore** Hessel et al., which reflects semantic consistency between generated images and input condition.

### B.1  REG VARIANTS AND COMPARISON TO OTHER BASELINES

We apply REG to three different parameterizations: the noise prediction $\epsilon$, the denoised prediction $D$, and the velocity $u$. Across all variants, we observe similar performance, demonstrating that REG is robust to the choice of application point. Qualitative results for REG variants are in Fig. 28 and Fig. 29.

FDG applies guidance in the image domain by leveraging a Laplacian pyramid, which limits its applicability to certain architectures. APG, while also employing EMA, deviates from our approach by using a negative $\beta$ parameter and directly adding low-frequency components without any rectification, leading to both qualitatively and quantitatively different outcomes.

For all baseline guidance methods, we use the best-known settings reported in the original papers or official GitHub repositories when available. If such configurations are not publicly released, we perform a grid search over nine combinations centered around the best-known settings from related models, and we report the results using the configuration that yields the best performance according to our primary evaluation metric.

### B.2  SAMPLING CONFIGURATION

All experiments are conducted using $4 \times$ NVIDIA 6000 Ada GPUs. For EDM2-s, we adopt Heun's method with 100 sampling steps. For SDXL, we use the Euler method with 50 sampling steps. Stable Diffusion 3 (SD3) is also sampled using the Euler method, with a total of 28 steps. These settings follow standard or recommended practices for each respective model to ensure a fair comparison across methods.

### B.3  HYPERPARAMETERS

For REG, we used the following hyperparameters for each model: in EDM2 Karras et al. (2022), we set $w = 1.6$, $\beta = 0.9$, and $\gamma = 0.4$; in SDXL Podell et al. (2023), we set $w = 10$, $\beta = 0.9$, and $\gamma = 0.4$; and in SD3 Esser et al. (2024), we set $w = 15$, $\beta = 0.9$, and $\gamma = 0.4$. For other guidance baselines, we adopted either the best hyperparameter settings provided in their respective papers or official GitHub repositories, or, when such information was unavailable, we performed a local grid search around known good configurations from similar models. The hyperparameters that achieved the highest evaluation metrics were used in our comparisons. The hyperparameters for other baseline guidance methods are summarized in Table 2.

| | EDM2-s | EDM2-xxl | SDXL | SD3 |
|---|---|---|---|---|
| **REG** | $w = 1.6$ 
 $\gamma = 0.4$ | $w = 1.6$ 
 $\gamma = 0.4$ | $w = 10$ 
 $\gamma = 0.4$ | $w = 15$ 
 $\gamma = 0.4$ |
| **CFG** | $\underline{w = 1.4}$ | $\underline{w = 1.2}$ | $\underline{w = 7}$ | $\underline{w = 12}$ |
| **APG** | $\underline{w = 4.0}$ 
 $\underline{\beta = -0.75}$ 
 $\underline{\eta = 0.0}$ 
 $\underline{r = 2.5}$ | $\underline{w = 2.0}$ 
 $\underline{\beta = -0.75}$ 
 $\underline{\eta = 0.0}$ 
 $\underline{r = 2.5}$ | $\underline{w = 15.0}$ 
 $\underline{\beta = -0.50}$ 
 $\underline{\eta = 0.0}$ 
 $\underline{r = 15}$ | $\underline{w = 20.0}$ 
 $\underline{\beta = -0.50}$ 
 $\underline{\eta = 0.0}$ 
 $\underline{r = 15}$ |
| **FDG** | $\underline{w = 3.0}$ 
 $\underline{w_{\text{low}} = 1.0}$ 
 $\underline{w_{\text{high}} = 3.0}$ | $\underline{w = 2.0}$ 
 $\underline{w_{\text{low}} = 1.0}$ 
 $\underline{w_{\text{high}} = 2.0}$ | $\underline{w = 10.0}$ 
 $\underline{w_{\text{low}} = 5.0}$ 
 $\underline{w_{\text{high}} = 10.0}$ | $\underline{w = 7.0}$ 
 $\underline{w_{\text{low}} = 3.0}$ 
 $\underline{w_{\text{high}} = 7.0}$ |
| **IG** | $\underline{w = 2.1}$ 
 $\underline{\sigma_{\text{min}} = 0.28}$ 
 $\underline{\sigma_{\text{max}} = 2.90}$ | $\underline{w = 2.0}$ 
 $\underline{\sigma_{\text{min}} = 0.19}$ 
 $\underline{\sigma_{\text{max}} = 1.61}$ | $\underline{w = 16}$ 
 $\underline{\sigma_{\text{min}} = 0.28}$ 
 $\underline{\sigma_{\text{max}} = 5.42}$ | $\underline{w = 15}$ 
 $\underline{\sigma_{\text{min}} = 0.65}$ 
 $\underline{\sigma_{\text{max}} = 1.40}$ |
| **TCFG** | $w = 1.4$ | $w = 1.2$ | $\underline{w = 5.0}$ | $\underline{w = 7.0}$ |
| **CFG++** | $w = 0.3$ | $w = 0.3$ | $\underline{w = 0.3}$ | $\underline{w = 0.1}$ |
| **AG** | $\underline{w = 2.10}$ | $\underline{w = 2.05}$ | - | - |

Table 2: Hyperparameters for baseline methods.

## C IMPLEMENTATION DETAILS

```python
class EMA:
    def __init__(
        self,
        m=0,
        beta=0.9,
        epsilon=1e-8
    ):
        self.m = m
        self.beta = beta
        self.epsilon = epsilon

    def update(self, Dx):
        self.m = self.beta * self.m + Dx
        return self.m

    def scale(self, v1, v2):
        v1_norm = torch.norm(
            v1,
            dim=1,
            keepdim=True
        )
        v2_norm = torch.norm(
            v2,
            dim=1,
            keepdim=True
        )
        scale_factor = v2_norm / \
            (v1_norm + self.epsilon)

        return scale_factor * v1

    def __call__(
        self,
        Dx,
```

```
        norm_scale=True,
    ):
        m_hat = self.update(Dx)

        if norm_scale:
            m_hat = self.scale(m_hat, Dx)

        return m_hat
```

## D  SIGNAL PROCESSING PERSPECTIVE ON THE REVERSE PROCESS

This section presents a simplified analytical model, grounded in signal processing principles, to elucidate the effects of temporal smoothing on the reverse process. While a rigorous analysis of the non-linear model like U-Net dynamics is presently intractable, this model provides a key insight. By modeling the coarse-to-fine nature of the generation process, we demonstrate that a temporal low-pass filter disproportionately preserves high spatial frequencies relative to low spatial frequencies.

### D.1  PREMISE FOR THE DIFFUSION MODEL REVERSE PROCESS: COARSE-TO-FINE DYNAMICS

We model the evolution of each spatial mode $k$ as a temporal signal, characterized by an effective rate of change, $\lambda_k$, averaged over the reverse process. This analytical framing highlights a fundamental asymmetry between the forward and reverse processes. In the standard forward diffusion process, white Gaussian noise is progressively added. While the noise itself has a flat power spectrum, its impact on natural images, which typically exhibit a $\frac{1}{f}$ power-law spectrum, is frequency-dependent. High-frequency components, having inherently lower initial power, are overwhelmed by noise more rapidly than their low-frequency counterparts. Thus, their signal-to-noise ratio degrades faster. Conversely, the diffusion model reverse process is widely understood to operate on a coarse-to-fine synthesis principle Park et al. (2023), an emergent property of its hierarchical architecture.

- **Low Spatial Frequencies** ($k_{low}$): These components define the image's global structure. They are established in the early-to-mid stages of the reverse process, where the image undergoes its most dramatic transformation from noise to a coherent form. This period of intense, large-scale change means that, when averaged over the entire process, low-frequency components exhibit a very high rate of temporal change ($\lambda_{k_{low}}$ is large).
- **High Spatial Frequencies** ($k_{high}$): These components represent fine details. They are synthesized and refined primarily in the later, more stable stages. Their synthesis is more incremental. Therefore, their average rate of temporal change is lower ($\lambda_{k_{high}}$ is small).

Thus, for our model to be applicable to the diffusion model, we must adopt a premise that is inverse to the forward process intuition: an increase in spatial frequency $k$ corresponds to a decrease in the effective rate of temporal change $\lambda$.

### D.2  EMA AS A TEMPORAL LOW-PASS FILTER

The EMA update rule, defined in this paper, $m_n = \beta m_{n-1} + \epsilon_n$ (where $n$ indexes discrete time steps) acts as a temporal low-pass filter (LPF). Li et al. (2024) Assuming a sampling interval $\Delta\tau$ for the temporal evolution of $\hat{G}_\tau(k)$, its frequency response $H_F(\omega_t)$ in the temporal frequency domain (where $\omega_t$ is the temporal angular frequency) is:

$$H_F(\omega_t) = \frac{1}{1 - \beta e^{-j\omega_t \Delta\tau}} \tag{8}$$

The magnitude $|H_F(\omega_t)|$ decreases as $|\omega_t|$ increases. For simplicity, we assume $\Delta\tau = 1$.

### D.3  TEMPORAL FREQUENCY CONTENT OF EVOLVING SPATIAL MODES

For a fixed spatial frequency $k$, we model the evolution of its corresponding component, $\hat{G}_\tau(k)$, as a stochastic process. The temporal dynamics of this process are governed by its characteristic

rate of change, $\lambda_k$. An analytically tractable approach for such processes is to assume that their temporal autocorrelation function decays exponentially at this rate Uhlenbeck & Ornstein (1930) Crispin (2009). By the Wiener-Khinchin theorem, the power spectral density (PSD) of the signal is the Fourier transform of its autocorrelation function. An exponential decay in the time domain corresponds to a Lorentzian profile in the frequency domain. We therefore model the PSD of the temporal signal for mode $k$ as:

$$|\hat{G}_k(\omega_t)|^2 = |A(k)|^2 \frac{1}{\lambda_k^2 + \omega_t^2} \tag{9}$$

Here, $|\hat{G}_k(\omega_t)|^2$ is shorthand for the PSD, and $|A(k)|^2$ is a frequency-dependent amplitude term. A larger $\lambda_k$ signifies a more rapid temporal evolution, which shifts the power distribution of the signal towards higher temporal frequencies $\omega_t$.

### D.4 EFFECT OF TEMPORAL LOW-PASS FILTERING ON SPATIAL FREQUENCIES

Applying the temporal LPF $H_F(\omega_t)$ to $\hat{G}_\tau(k)$, the energy of the filtered signal, $E_k^{\text{after}}$, is:

$$E_k^{\text{after}} = \frac{1}{2\pi} \int_{-\infty}^{\infty} |H_F(\omega_t)|^2 |\hat{G}_k(\omega_t)|^2 d\omega_t$$
$$= \frac{|A(k)|^2}{2\pi} \int_{-\infty}^{\infty} |H_F(\omega_t)|^2 \frac{1}{\lambda_k^2 + \omega_t^2} d\omega_t \tag{10}$$

The energy retention ratio $R_k = E_k^{\text{after}}/E_k^{\text{before}}$ is:

$$R_k = \frac{\lambda_k}{\pi} \int_{-\infty}^{\infty} |H_F(\omega_t)|^2 \frac{1}{\lambda_k^2 + \omega_t^2} d\omega_t$$

Using the substitution $\omega_t = \lambda_k \nu$:

$$R_k = \frac{1}{\pi} \int_{-\infty}^{\infty} |H_F(\lambda_k \nu)|^2 \frac{1}{1 + \nu^2} d\nu \tag{11}$$

We can now analyze this retention ratio using the coarse-to-fine premise established for the diffusion reverse process. Consider two spatial frequencies $k_1$ and $k_2$ with $\|k_2\| > \|k_1\|$. According to our premise, their corresponding temporal change rates satisfy $\lambda_{k_1} > \lambda_{k_2}$. Since $|H_F(\omega_t)|$ is non-increasing function of $|\omega_t|$ (as it is an LPF) and $\lambda_{k_1} > \lambda_{k_2} > 0$, for $\nu \neq 0$, we have $|\lambda_{k_1} \nu| > |\lambda_{k_2} \nu|$. Consequently, the filter response magnitude satisfies $|H_F(\lambda_{k_1} \nu)|^2 \leq |H_F(\lambda_{k_2} \nu)|^2$. As this inequality holds over the domain of integration, the integral in Eq. 11 will be smaller for $k_1$ than for $k_2$. This leads to the central result:

$$\int_{-\infty}^{\infty} |H_F(\lambda_{k_1} \nu)|^2 \frac{1}{1 + \nu^2} d\nu < \int_{-\infty}^{\infty} |H_F(\lambda_{k_2} \nu)|^2 \frac{1}{1 + \nu^2} d\nu$$

(assuming $H_F$ is not constant). Therefore:

$$R_{k_1} < R_{k_2} \quad \text{for } \|k_2\| > \|k_1\| \tag{12}$$

Inequality Eq. 12 establishes a key theoretical prediction: the temporal low-pass filtering inherent in EMA has a stronger attenuating effect on low spatial frequencies than on high spatial frequencies. This result, derived from our analytical framework and the coarse-to-fine premise of diffusion model synthesis, aligns with empirical observations of improved sample quality. The model suggests that temporal smoothing enhances generation not by introducing blur, but by selectively damping the large, rapid fluctuations associated with the formation of coarse image structures, thereby allowing the more stable, incremental synthesis of fine details to proceed with less interference.

## E RELATED WORK

Recent work on guidance for diffusion models can be broadly categorized into training-free and training-based approaches. Within training-free methods, architecture-dependent techniques such as PAG Ahn et al. (2024), SAG Hong et al. (2023), SEG Hong (2024), and STG Hyung et al. (2024)

modify internal blocks to form a weak reference; while effective, reliance on model internals limits portability. A second strand is image-dependent guidance, where rules adapt to sample properties: DNG Koulischer et al. (2025) schedules negative prompting, and FDG Sadat et al. (2025) decouples frequencies to raise fidelity. Although FDG shares a conceptual similarity with our method (REG) in connecting sample diversity to low-frequency components, they diverge in implementation: FDG explicitly manipulates spatial frequencies via Laplacian pyramid decomposition, whereas REG operates on the temporal dynamics of the model's predictions via EMA without requiring explicit frequency decomposition.

The third—and most broadly applicable—strand is architecture-independent, training-free guidance (e.g., ADG Jin et al. (2025), TCFG Kwon et al. (2025), CFG++ Chung et al. (2024), CADS Sadat et al. (2024a)). Adaptive Projected Guidance (APG) Sadat et al. (2024b) also falls into this category and utilizes an Exponential Moving Average (EMA) to mitigate oversaturation. However, its mechanism differs fundamentally from ours: APG employs a negative momentum parameter ($\beta < 0$) to act as a step-wise cancellation signal, whereas REG uses its EMA-derived term to rectify the guidance signal itself. This distinction allows REG to extend stability up to higher guidance scales (e.g., 20 on SDXL) compared to APG (limit of 15). To address the entanglement of condition strength and perceptual quality in these methods, other timestep-oriented fixes have been proposed: IG Kynkäänniemi et al. (2024) applies guidance only in a limited noise interval, ReCFG Sadat et al. (2024a) rectifies coefficients to remove expectation shift, and ICG/TSG Sadat et al. (2024c) introduces a time-shifted schedule.

Finally, training-based approaches use an auxiliary model: AG Karras et al. (2024a) learns or provisions a weaker "bad" model and guides by the strong–weak difference to improve quality without collapsing diversity, while PG Park et al. (2025) steers a fine-tuned personalized model toward a balanced space via weight interpolation—both effective but dependent on extra training/fine-tuning. Beyond these, several works explicitly distill classifier-free guidance into faster or modified generators. While necessitating additional training, several studies have attempted to use the guidance scale as a condition during model learning. One line of research encodes this condition into the generator parameters: Meng et al. (2023) compress the CFG process into a student model, and Dufour et al. (2025) train the model to leverage a label-coherence score. In contrast, Hsiao et al. (2024) handles the guidance condition externally by learning an auxiliary network to approximate CFG updates. Our method differs fundamentally as it is purely training-free, avoiding the need for any such optimization.

# F    UNIFIED PARAMETERIZATION OF PREDICTION TARGETS IN GAUSSIAN PATH MODELS

We provide a unified formulation of prediction targets - noise ($\epsilon$), denoised sample ($D$), and velocity ($u$)—under the Gaussian path framework and its applications to diffusion and flow-based models.

**Gaussian Paths and Schedulers**    Let $p_t(x \mid x_1) = \mathcal{N}(x; \alpha_t x, \sigma_t^2 I)$ be a Gaussian path, where $x_0 \sim \mathcal{N}(0, I), x_1 \sim p_1$, and $x_t = \alpha_t x_1 + \sigma_t x_0$. The function $\alpha_t, \sigma_t \colon [0, 1] \to [0, 1]$ are continuously differentiable and monotonic, that form a scheduler, satisfying boundary conditions: $\alpha_0 = \sigma_1 = 0$ and $\alpha_1 = \sigma_0 = 1$. We focus on three representative choices of schedulers commonly used in continuous-time generative modeling:

- Variance-Preserving (VP) Ho et al. (2020): $\alpha_t^2 + \sigma_t^2 = 1$

- EDM Karras et al. (2022): $\alpha_t = 1, \sigma_t = t$

- Rectified Flow Liu et al. (2022): $\alpha_t = 1, \sigma_t = 1 - t$

**Conversion between Velocity Field and Score**    Let $p_t(x \mid y)$ be a Gaussian path with scheduler $(\alpha_t, \sigma_t)$. Then the velocity field generating this path satisfies:

$$u_t(x \mid y) = a_t x + b_t \nabla_x \log p_t(x \mid y)$$

with $(a_t, b_t) = (\frac{\dot{\alpha}_t}{\alpha_t}, \frac{\dot{\alpha}_t \sigma_t^2 - \dot{\sigma}_t \sigma_t \alpha_t}{\alpha_t})$ by Eq. (7) in Zheng et al. (2023).

# G  2D FRACTAL-LIKE GAUSSIAN MIXTURE TOY DATASET

We use the 2D toy dataset introduced in AG Karras et al. (2024a), which defines a synthetic probability distribution over a 2D space using a fractal-like mixture of Gaussians. The distribution is constructed recursively: each branch spawns two sub-branches with slightly randomized orientations and lengths, and each branch is modeled using 8 anisotropic Gaussian components. This recursive process is repeated up to depth 7, resulting in 1,016 components per class.

The resulting density is both structurally complex and analytically tractable. It captures two key characteristics commonly attributed to real-world image manifolds: low local dimensionality, as the density is concentrated along narrow, anisotropic structures, and hierarchical detail, as finer features emerge gradually with decreasing noise levels. For visualization, we render log-probability contours over a $400 \times 400$ spatial grid. In particular, we highlight the contour at level -2.12, which approximately delineates high-density regions where $\log p(x) \gtrsim -2.12$.

When applying various training-free guidance methods to this dataset, we compared the best-case outcomes by sweeping over different guidance scales for each method and selecting the setting that resulted in the least amount of branch drop. Among all methods, only REG and AG successfully preserved the full branching structure, with REG achieving this without introducing any additional parameters (the weak version of itself). The visualization of toy distribution is in Fig. 19

**Scale and hyperparameters settings**   We summarize the scale ranges and hyperparameter configurations for each guidance method as follows.

- **CFG** : Guidance scale values increase from top to bottom, ranging from 2.0 to 5.0 in increments of 0.5.

- **APG** : Guidance scale values increase from top to bottom, ranging from 2.0 to 5.0 in increments of 0.5. beta scale 0.75. eta scale 0.3. r scale 1.8

- **IG** : Guidance scale values increase from top to bottom, ranging from 2.0 to 5.0 in increments of 0.5. sigma low scale 10. sigma high 20.

- **TCFG** : Guidance scale values increase from top to bottom, ranging from 2.0 to 5.0 in increments of 0.5.

- **CFG++** : Guidance scale values increase from top to bottom, ranging from 0.05 to 0.35 in increments of 0.05.

- **AG** : Guidance scale values increase from top to bottom, ranging from 2.0 to 5.0 in increments of 0.5.

- **REG** : Guidance scale values increase from top to bottom, ranging from 4.0 to 10.0 in increments of 1.0. Beta ($\beta$) scale 0.9. Gamma scale ($\gamma$) 1.0.

# H  FORMULATION COMPARISON WITH APG

In this section, we mathematically analyze the structural differences between Adaptive Projected Guidance (APG) and our proposed method (REG). By decomposing the update rules of both methods into comparable components, we highlight how REG offers a more direct and effective regularization compared to the geometric projections used in APG.

## H.1  APG (ADAPTIVE PROJECTED GUIDANCE)

APG introduces a EMA-based guidance term in the sampling process. Let $\hat{u}_{t_k}(x \mid y)$ and $\hat{u}_{t_k}(x)$ denote the conditional and unconditional predictions, respectively. With hyperparameters $\beta < 0$ and $\eta \leq 1$, the EMA term $\hat{u}_{t_k}^m$ is defined recursively as:

$$\hat{u}_{t_k}^m(x \mid y) = \beta \hat{u}_{t_{k-1}}^m(x \mid y) + s_{t_k}(\hat{u}_{t_k}(x \mid y) - \hat{u}_{t_k}(x))$$

$$= \sum_{i=0}^{k} \beta^i (s_{t_{k-i}}(\hat{u}_{t_{k-i}}(x \mid y) - \hat{u}_{t_{k-i}}(x))) \tag{13}$$

where $s_k = \min(1, \frac{\tau}{\|\hat{u}_{t_k}^m(x|y)\|})$ is a scaling factor with threshold $\tau$ to constrain the magnitude of the EMA. APG projects this EMA term $\hat{u}_{t_k}^m(x \mid y)$ onto the current conditional prediction $\hat{u}_{t_k}(x \mid y)$. This decomposition yields a parallel component, $\hat{u}_{t_k}^{\|}(x \mid y) = \frac{\langle s_{t_k}(\hat{u}_{t_k}(x|y) - \hat{u}_{t_k}(x)), \hat{u}_{t_k}(x|y)\rangle}{\langle \hat{u}_{t_k}(x|y), \hat{u}_{t_k}(x|y)\rangle} \hat{u}_{t_k}(x \mid y)$, and an orthogonal component, $\hat{u}_{t_k}^{\perp}(x \mid y) = \hat{u}_{t_k}^m(x \mid y) - \hat{u}_{t_k}^{\|}(x \mid y)$.

The final APG update rule combines these components. By expanding the terms, we can reformulate APG to reveal its underlying relationship with standard CFG:

$$
\begin{aligned}
\tilde{u}_{t_k}^{apg}(x \mid y) &= \hat{u}_{t_k}(x \mid y) + w(\hat{u}_{t_k}^{\perp}(x \mid y) + \eta\hat{u}_{t_k}^{\|}(x \mid y)) \\
&= \hat{u}_{t_k}(x \mid y) + w(\hat{u}_{t_k}^m(x \mid y) - \hat{u}_{t_k}^{\|}(x \mid y) + \eta\hat{u}_{t_k}^{\|}(x \mid y)) \\
&= \hat{u}_{t_k}(x \mid y) + w\hat{u}_{t_k}^m(x \mid y) - w(1 - \eta)\hat{u}_{t_k}^{\|}(x \mid y) \\
&= \hat{u}_{t_k}(x \mid y) + w(\beta\hat{u}_{t_{k-1}}^m(x \mid y) + s_k(\hat{u}_{t_k}(x \mid y) - \hat{u}_{t_k}(x))) - w(1 - \eta)\hat{u}_{t_k}^{\|}(x \mid y) \\
&= \hat{u}_{t_k}(x \mid y) + ws_{t_k}(\hat{u}_{t_k}(x \mid y) - \hat{u}_{t_k}(x)) + w\beta\hat{u}_{t_{k-1}}^m(x \mid y) - w(1 - \eta)\hat{u}_{t_k}^{\|}(x \mid y) \\
&= \hat{u}_{t_k}(x \mid y) + ws_{t_k}(\hat{u}_{t_k}(x \mid y) - \hat{u}_{t_k}(x)) + w(\beta\hat{u}_{t_{k-1}}^m(x \mid y) - (1 - \eta)\hat{u}_{t_k}^{\|}(x \mid y)) \\
&= \hat{u}_{t_k}(x \mid y) + ws_{t_k}(\hat{u}_{t_k}(x \mid y) - \hat{u}_{t_k}(x)) \\
&\quad + w\left(\beta\sum_{i=0}^{k-1}\beta^i(s_{t_{k-1-i}}(\hat{u}_{t_{k-1-i}}(x \mid y) - \hat{u}_{t_{k-1-i}}(x))) - (1 - \eta)\hat{u}_{t_k}^{\|}(x \mid y)\right) \\
&= \hat{u}_{t_k}(x \mid y) + ws_k(\hat{u}_{t_k}(x \mid y) - \hat{u}_{t_k}(x)) \\
&\quad + w\beta\left(\sum_{i=0}^{k-1}\beta^i(s_{t_{k-1-i}}(\hat{u}_{t_{k-1-i}}(x \mid y) - \hat{u}_{t_{k-1-i}}(x))) - (1 - \eta)\hat{u}_{t_k}^{\|}(x \mid y)\right) \\
&= \hat{u}_{t_k}(x \mid y) + \underbrace{ws_{t_k}(\hat{u}_{t_k}(x \mid y) - \hat{u}_{t_k}(x))}_{\text{Down-scaled CFG boosting term}} \\
&\quad + w\beta\underbrace{\left(\sum_{i=0}^{k-1}\beta^i(s_{t_{k-1-i}}(\hat{u}_{t_{k-1-i}}(x \mid y))) - \sum_{i=0}^{k-1}\beta^i(s_{t_{k-1-i}}(\hat{u}_{t_{k-1-i}}(x)))\right)}_{\text{Reverse EMA conditional term}} \\
&\quad - \underbrace{w(1 - \eta)\hat{u}_{t_k}^{\|}(x \mid y)}_{\text{Parallel conditional term}}
\end{aligned}
$$
(14)

Based on the decomposition above, we analyze the distinct role of each term in APG:

- **Down-scaled CFG boosting term:** This term acts as the primary guidance force, containing both the *conditional component* and the *quality component*. In APG, this term is scaled down by a factor $s_k$ to uniformly lower the guidance influence.

- **Reverse EMA conditional term:** This term modulates the guidance using historical statistics. By applying the scaling factor $s$ and utilizing a negative decay rate ($\beta < 0$), the EMA self-cancels at every step. Consequently, instead of smoothing the trajectory, this *reverse* mechanism functions to boost the variation of the model prediction at the current step. It emphasizes immediate changes rather than long-term trends.

- **Parallel conditional term:** This term functions as a correction mechanism. Within the accumulated reverse EMA, it operates to cancel out components parallel to the combined conditional and quality direction. Since this term aligns with the coupled direction, it is structurally limited in isolating the conditional component, which may lead to insufficient suppression of the accumulated conditional influence at high guidance scales.

## H.2 REG

Let $\hat{u}_{t_k}(x \mid y)$ and $\hat{u}_{t_k}(x)$ denote the conditional and unconditional predictions, respectively. With hyperparameters $\beta = 0.9$ and $\gamma \in (0.3, 0.6)$, the conditional and unconditional EMA terms, $\hat{u}_{t_k}^m(x \mid y)$ and $\hat{u}_{t_k}^m(x)$, are defined recursively as:

$$\hat{u}_{t_k}^m(x \mid y) := \sum_{s=0}^{k} \beta^s \hat{u}_{t_{k-s}}(x \mid y) \qquad (15) \qquad\qquad \hat{u}_{t_k}^m(x) := \sum_{s=0}^{k} \beta^s \hat{u}_{t_{k-s}}(x) \qquad (16)$$

In addition, the scaling terms are defined as $s_{t_k}^c = \frac{\|\hat{u}_{t_k}(x|y)\|}{\|\hat{u}_{t_k}^m(x|y)\|}$ and $s_{t_k}^u = \frac{\|\hat{u}_{t_k}(x)\|}{\|\hat{u}_{t_k}^m(x)\|}$.

In contrast to the geometric projection of APG, REG simplifies the stabilization process by directly minimizing the conditional component. The REG update rule is derived as follows:

$$
\begin{aligned}
\tilde{u}_{t_k}^{reg}(x \mid y) &= (1+w)\hat{u}_{t_k}(x \mid y) - w(\hat{u}_{t_k}(x) + \gamma(s_{t_k}^c \hat{u}_{t_k}^m(x \mid y) - s_{t_k}^u \hat{u}_{t_k}^m(x))) \\
&= (1+w)\hat{u}_{t_k}(x \mid y) - w\hat{u}_{t_k}(x) - w\gamma(s_{t_k}^c \hat{u}_{t_k}^m(x \mid y) - s_{t_k}^u \hat{u}_{t_k}^m(x)) \\
&= \hat{u}_{t_k}(x \mid y) + w(\hat{u}_{t_k}(x \mid y) - \hat{u}_{t_k}(x)) - w\gamma(s_{t_k}^c \hat{u}_{t_k}^m(x \mid y) - s_{t_k}^u \hat{u}_{t_k}^m(x)) \\
&= \hat{u}_{t_k}(x \mid y) + w(\hat{u}_{t_k}(x \mid y) - \hat{u}_{t_k}(x)) \\
&\quad - w\gamma\left(s_{t_k}^c \sum_{s=0}^{k} \beta^s(\hat{u}_{t_{k-s}}(x \mid y)) - s_{t_k}^u \sum_{s=0}^{k} \beta^s(\hat{u}_{t_{k-s}}(x))\right) \\
&= \hat{u}_{t_k}(x \mid y) + \underbrace{w(\hat{u}_{t_k}(x \mid y) - \hat{u}_{t_k}(x))}_{\text{CFG boosting term}} \\
&\quad - \underbrace{w\gamma\left(s_{t_k}^c \sum_{s=0}^{k} \beta^s(\hat{u}_{t_{k-s}}(x \mid y)) - s_{t_k}^u \sum_{s=0}^{k} \beta^s(\hat{u}_{t_{k-s}}(x))\right)}_{\text{EMA conditional term}}
\end{aligned}
\qquad (17)
$$

As shown in the equation, REG cleanly separates the guidance mechanism into two distinct parts:

- **CFG boosting term:** This term remains identical to the original CFG formulation, containing both the *conditional component* and the *quality component* without modification.

- **EMA conditional term:** This term leverages the Low-Pass Filter (LPF) characteristic of standard EMA while preserving the original CFG structure. By subtracting the smoothed historical predictions, it effectively cancels out the *conditional component*, which tends to be low-frequency, from the guidance signal. This subtraction allows the method to primarily enhance the *quality component*, enabling stable sampling even at high guidance scales.

## I  ADDITIONAL EXPERIMENTAL RESULTS

**Ablation Study.**  We analyze the effect of two key hyperparameters in our method: the gamma scale ($\gamma$) and the beta scales ($\beta_c$, $\beta_u$), which independently modulate the conditional and unconditional EMA components. While the main paper denotes the EMA decay parameter simply as $\beta$ for conceptual clarity, in this appendix we decompose it into $\beta_c$ (conditional decay scale) and $\beta_u$ (unconditional decay scale) to examine their distinct effects.

In Fig. 12, $\gamma$ is varied from 0.0 to 1.0. Empirically, image quality is highest when $\gamma$ is set to 0.4 or 0.6. For $\gamma < 0.4$, the improvement is marginal, while values greater than 0.6 result in degraded image quality and structural collapse. Additional result is in Fig. 25

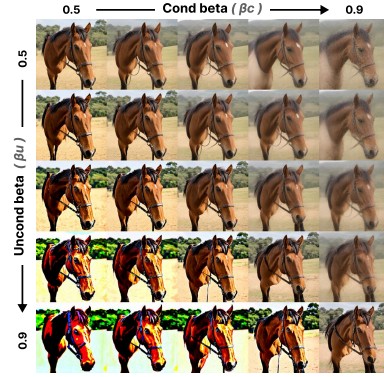

Figure 10: **Beta scale ($\beta_c$, $\beta_u$) comparison.** Guidance scale $w=9$ and $\gamma=0.6$.

**Increasing Guidance Scales** ⟶

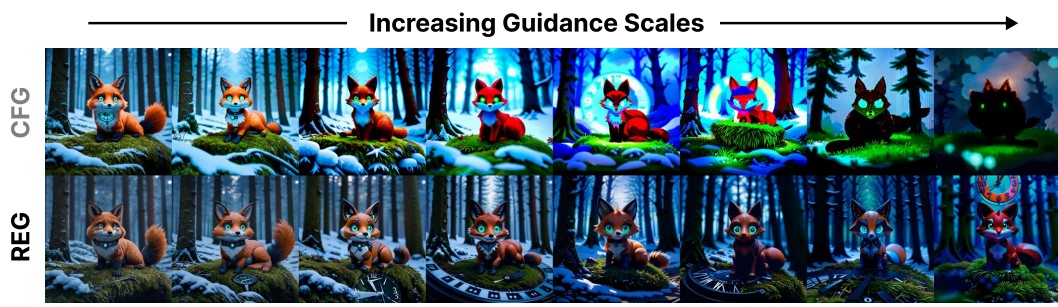

Figure 11: **Guidance scale ($w$) comparison**. Comparison of CFG (top) and REG (bottom) with SD3 across guidance scales ($w$ = 5 to 40, step 5). CFG yields oversimplified and over-saturated results as $w$ increases, while REG preserves structural detail and consistently enhances image quality. Prompt: *"A mechanical fox with glowing cyan eyes lounges on a moss-covered clock face in a forest frozen mid-autumn."*

| SD3 | $w$ | 3 | 6 | 9 | 12 | 15 |
|---|---|---|---|---|---|---|
| **CFG** | **FID** | 29.24 | 30.18 | 28.98 | 28.50 | 33.07 |
| | **CLIPScore** | 31.48 | 31.57 | 31.55 | 31.51 | 31.40 |
| | **Precision** | 0.50 | 0.61 | 0.59 | 0.55 | 0.47 |
| | **Recall** | 0.41 | 0.36 | 0.31 | 0.25 | 0.21 |
| **REG** | **FID** | 29.06 | 29.68 | 29.79 | 28.87 | 27.66 |
| | **CLIPScore** | 31.37 | 31.41 | 31.50 | 31.45 | 31.43 |
| | **Precision** | 0.43 | 0.60 | 0.60 | 0.60 | 0.57 |
| | **Recall** | 0.42 | 0.40 | 0.39 | 0.38 | 0.34 |

Table 3: **Guidance scale ($w$) sweep on SD3.** Quantitative comparison of **CFG** vs **REG** across $w \in \{3, 6, 9, 12, 15\}$ . As $w$ increases, CFG results tend to oversimplify structure and over-saturate, while REG at matched $w$ preserves structural detail and maintains higher visual quality.

| SDXL | $w$ | 7 | 10 | 13 | 18 |
|---|---|---|---|---|---|
| **CFG** | **FID** | 27.43 | 27.61 | 27.92 | 29.27 |
| | **CLIPScore** | 31.93 | 32.04 | 32.08 | 32.04 |
| | **Precision** | 0.56 | 0.59 | 0.61 | 0.59 |
| | **Recall** | 0.35 | 0.32 | 0.29 | 0.26 |
| **REG** | **FID** | 27.67 | 27.05 | 27.26 | 27.60 |
| | **CLIPScore** | 31.79 | 32.00 | 32.08 | 32.11 |
| | **Precision** | 0.54 | 0.57 | 0.60 | 0.61 |
| | **Recall** | 0.37 | 0.35 | 0.33 | 0.33 |

Table 4: **Guidance scale ($w$) sweep on SDXL.** Quantitative comparison of **CFG** vs **REG** across $w \in \{7, 10, 13, 18\}$.

In Fig. 10, $\beta_c$ and $\beta_u$ are varied from 0.5 to 0.9. It is observed that when $\beta_c > \beta_u$, the generated images tend to appear blurry. When $\beta_c = \beta_u$, the results are most visually appealing, whereas $\beta_c < \beta_u$ often causes oversaturation. Therefore, we set both $\beta_c$ and $\beta_u$ to 0.9 in all qualitative evaluations. For completeness, we additionally provide extended visual examples in Fig. 26.

**Various sampling step analysis.** To examine how the sampling steps affect generative stability, we visualize results in Fig. 27 using SDXL and SD3 across steps ranging from 50 to 100 in increments of 10. As shown, increasing the number of sampling steps does not degrade image quality, indicating that both models remain robust even at larger sampling step settings.

**Increasing γ Scales** ➝

Figure 12: **Gamma scale (γ) comparison**. Ablation study of γ scale with SD3 at guidance scales $w = 9$. REG scales from 0.0 to 1.0 (step 0.2).

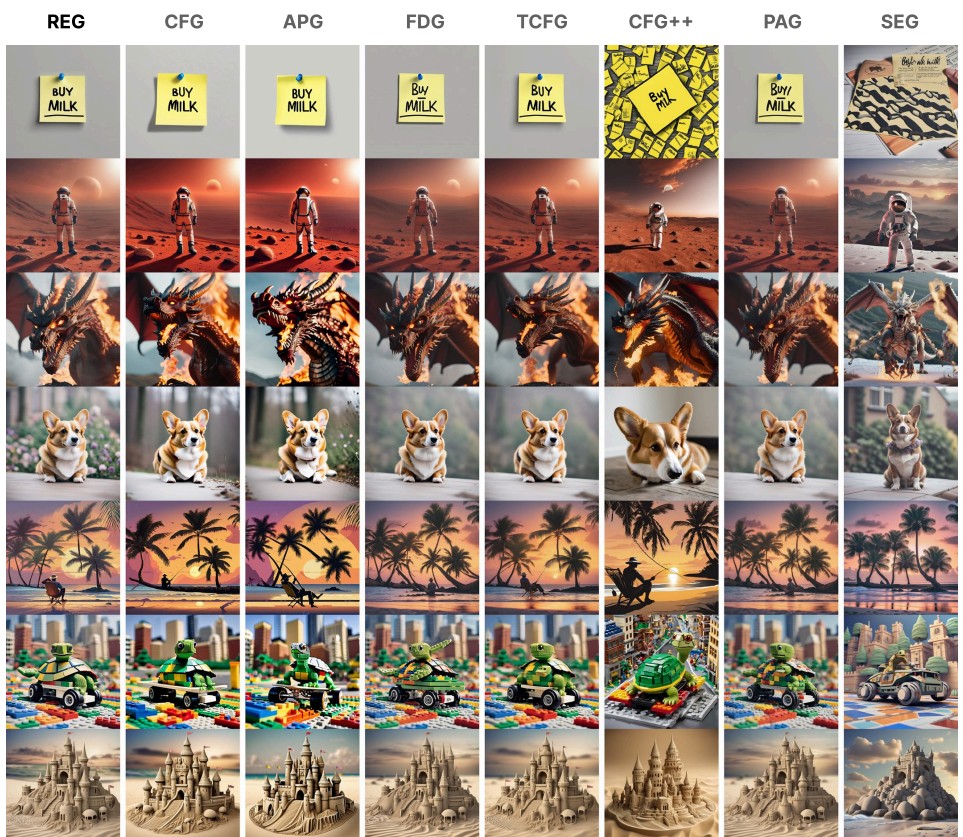

Figure 13: **Qualitative comparison of guidance methods on SDXL.**

| SDXL | REG | CFG | APG | FDG | TCFG | CFG++ | PAG | SEG |
|---|---|---|---|---|---|---|---|---|
| **FID** (↓) | 27.05 | 27.22 | 25.83 | 28.45 | 29.53 | 25.13 | 31.04 | 39.87 |
| **CLIPScore** (↑) | 32.00 | 31.94 | 32.31 | 32.07 | 31.32 | 31.76 | 31.17 | 29.65 |
| **Precision** (↑) | 0.57 | 0.61 | 0.54 | 0.57 | 0.48 | 0.68 | 0.56 | 0.37 |
| **Recall** (↑) | 0.35 | 0.35 | 0.35 | 0.35 | 0.37 | 0.29 | 0.41 | 0.23 |

Table 5: **Quantitative comparison of guidance methods on SDXL.**

REG CFG APG FDG IG TCFG CFG++

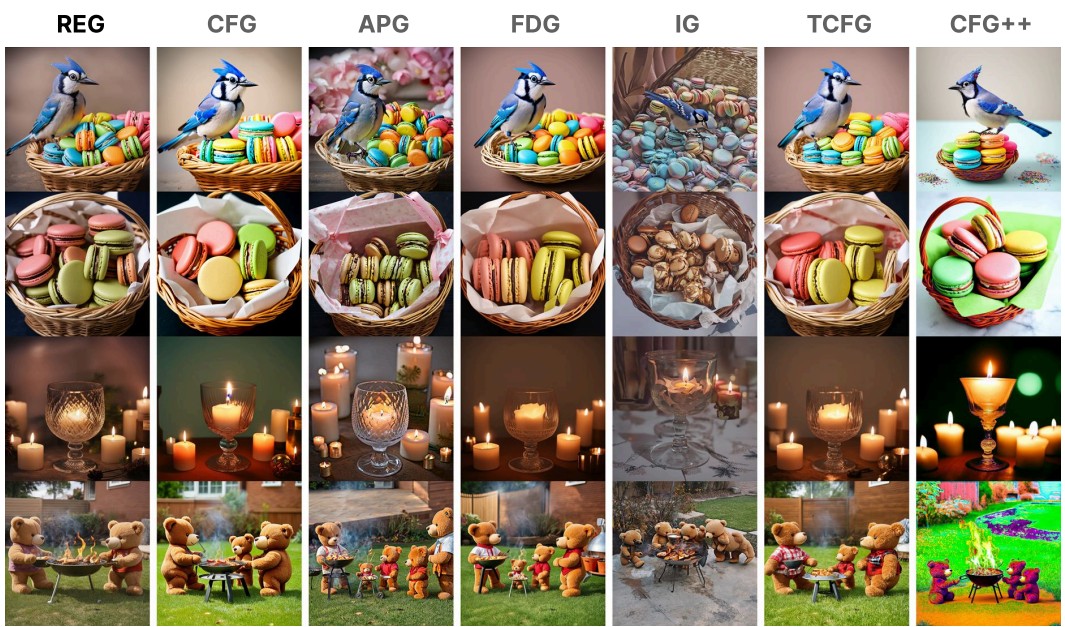

Figure 14: **Whole qualitative comparison with SD3**. Qualitative results generated using all models compared in Table 1. Prompt : *"A blue jay standing on a large basket of rainbow macarons.", "A basket of macarons", "A candle is lit in a large glass goblet with other candles around it", "a family of teddy bears having a barbecue in their backyard."*

**" beautiful lady, freckles, big smile, blue eyes, short ginger hair, wearing a floral blue vest top, soft light, dark gray background "**

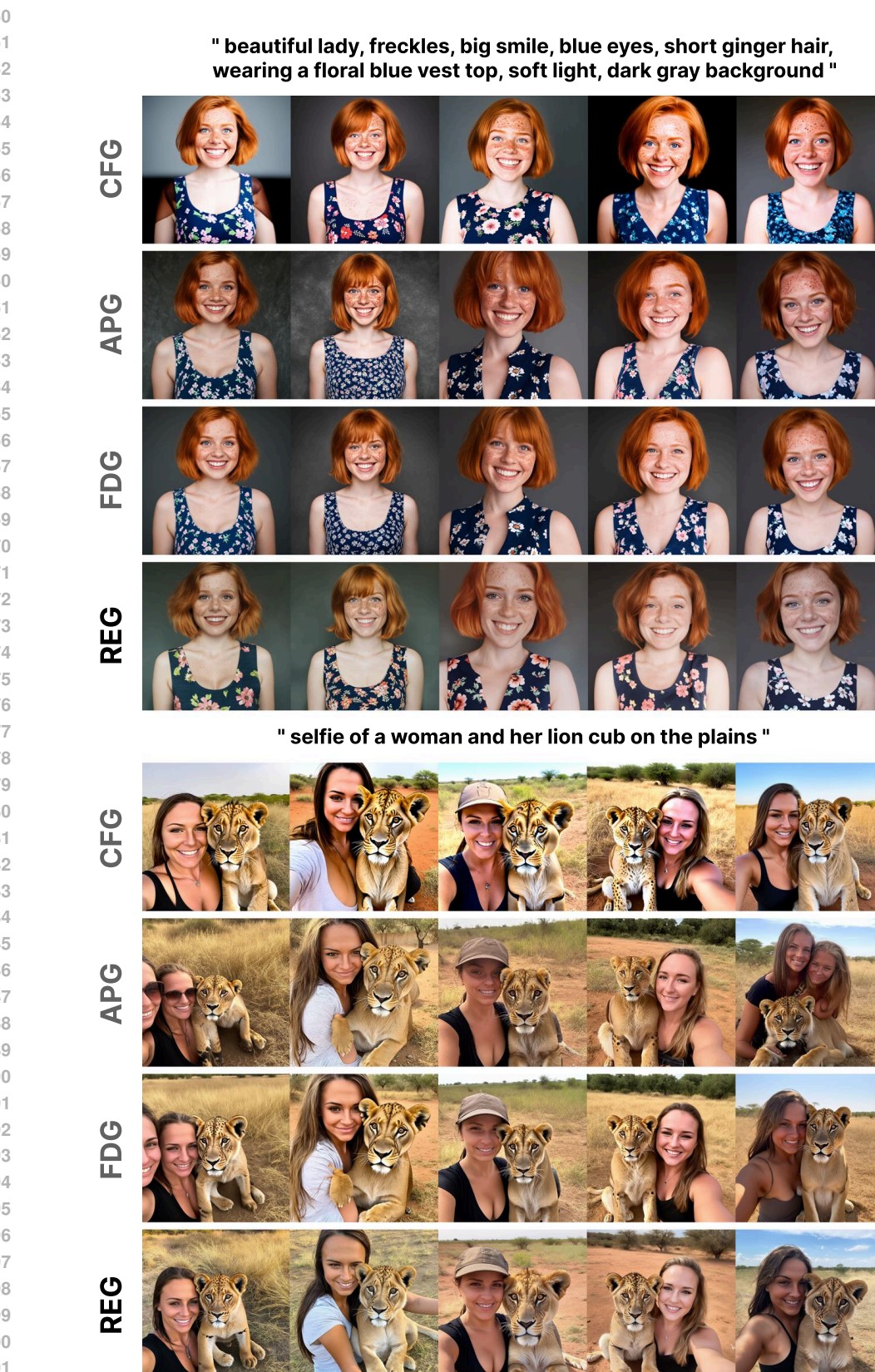

**" selfie of a woman and her lion cub on the plains "**

Figure 15: **Additional qualitative comparison with SD3**

**Increasing Guidance Scales** →

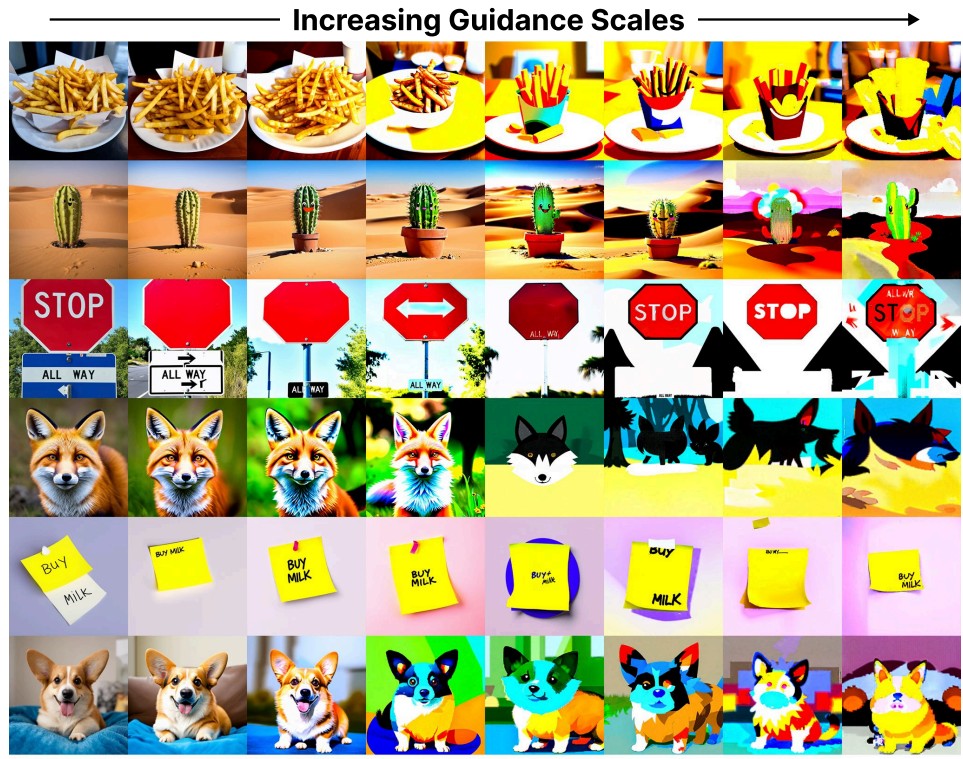

**(a) CFG**

**Increasing Guidance Scales** →

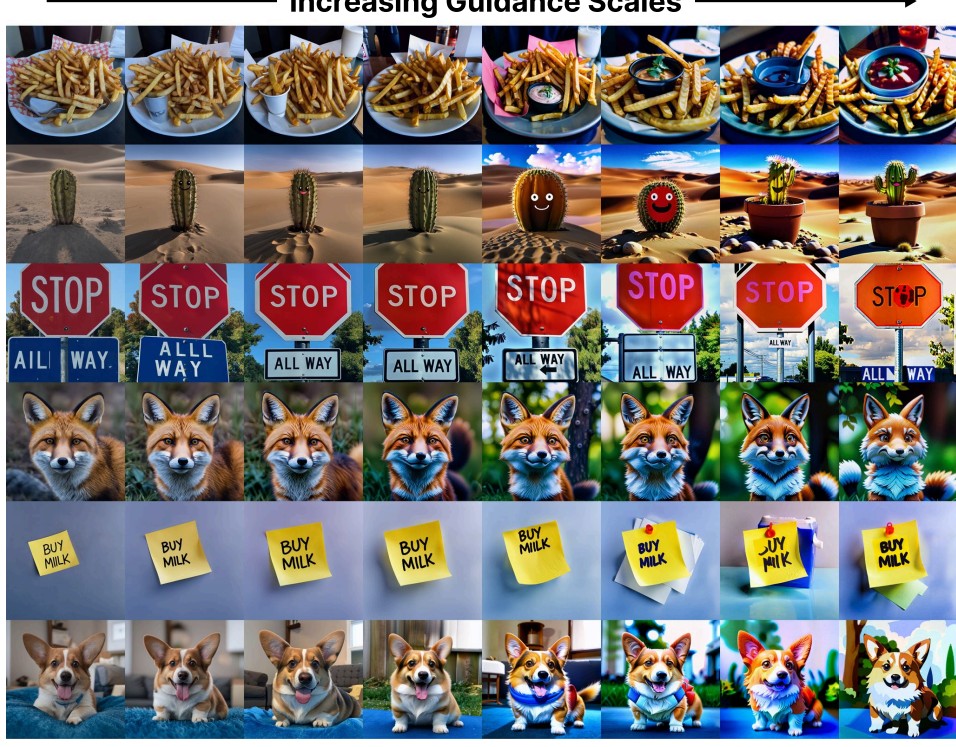

**(b) REG**

Figure 16: **Additional guidance scale ($w$) comparison with SD3**. Additional result from Figure 11. Prompt : *"A plate of fries", "A small cactus with a happy face in the Sahara desert", "A stop sign with "ALL WAY" written below it", "A fox", "A yellow sticky note with "BUY MILK" written on it", "Image of a corgi"*

**(a) CFG**

**(b) REG**

Figure 17: **Additional guidance scale ($w$) comparison with SDXL**. Prompt : *"A pink dog", "A 4k dslr photo of a raccoon wearing an astronaut helmet, photorealistic", "A golden retriever", "Astronaut on Mars during sunset", "A statue of a greek man"*

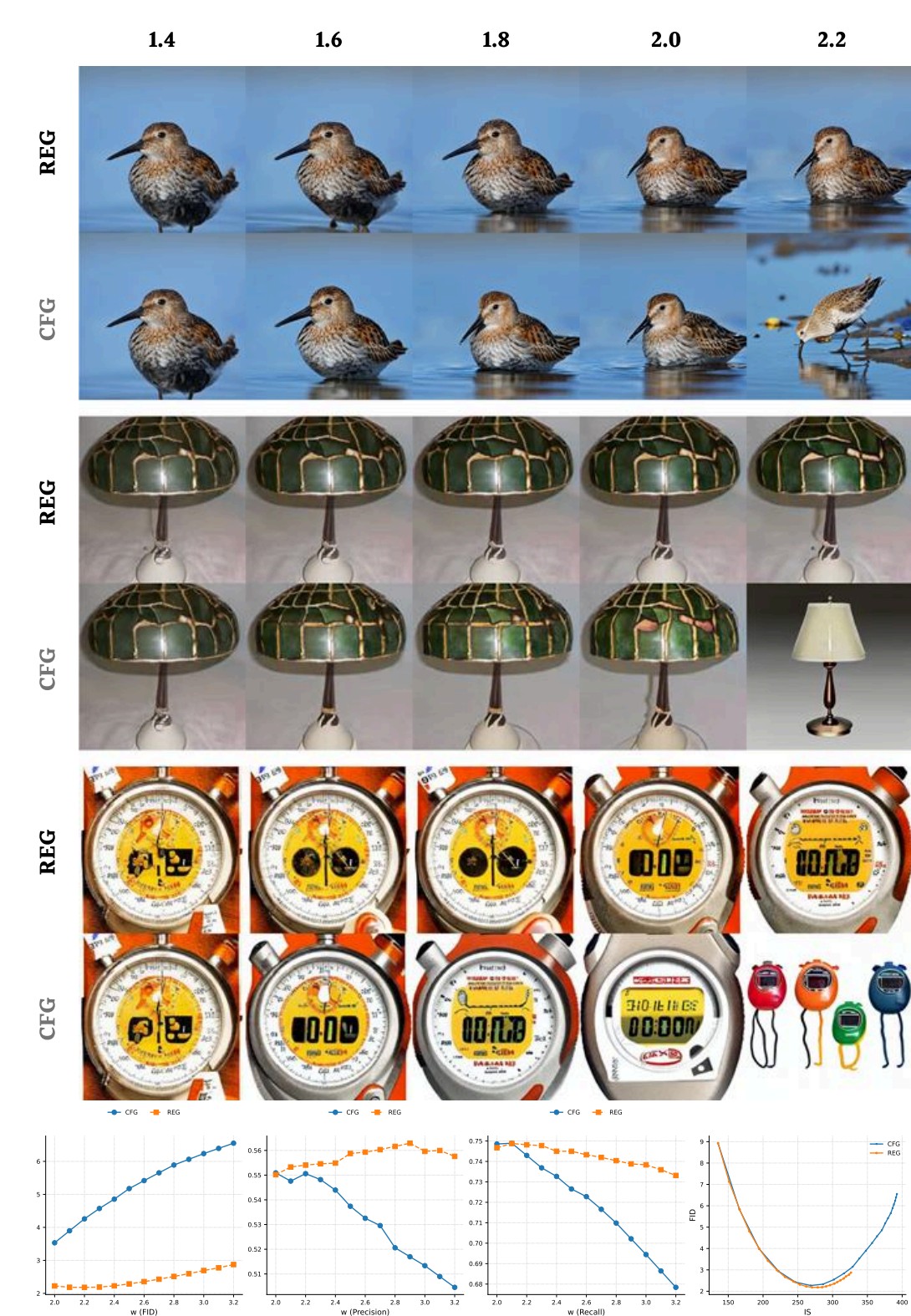

Figure 18: **The diversity-preserving effect of REG on EDM2-s.** REG maintains image quality even as the guidance scale increases.

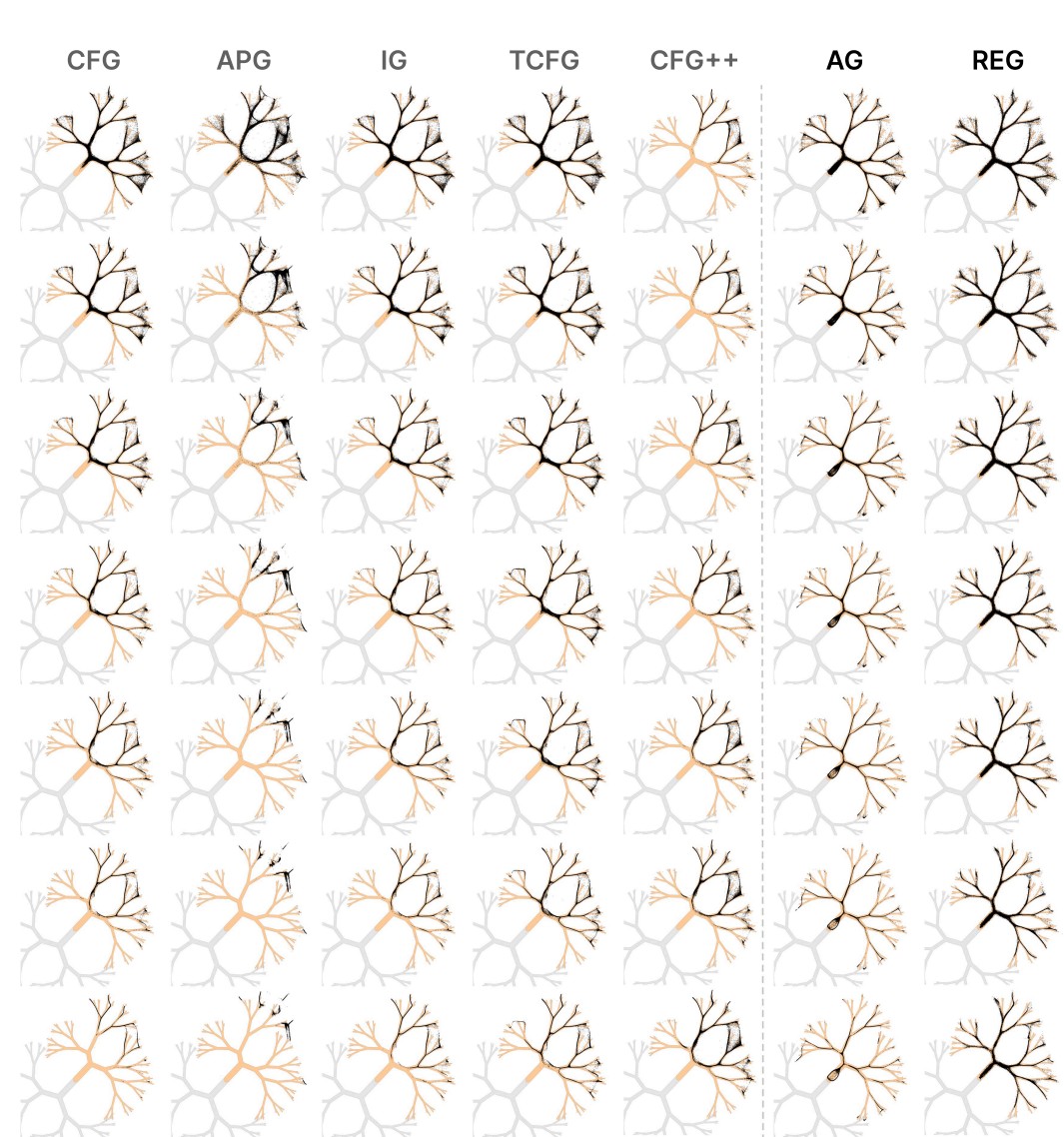

Figure 19: **Diversity preserving property of REG in fractal-like 2D distribution**. Only REG and AG Karras et al. (2024a) are able to preserve diversity in high guidance scale, while, REG does not requires additional parameters (the bad version of itself).

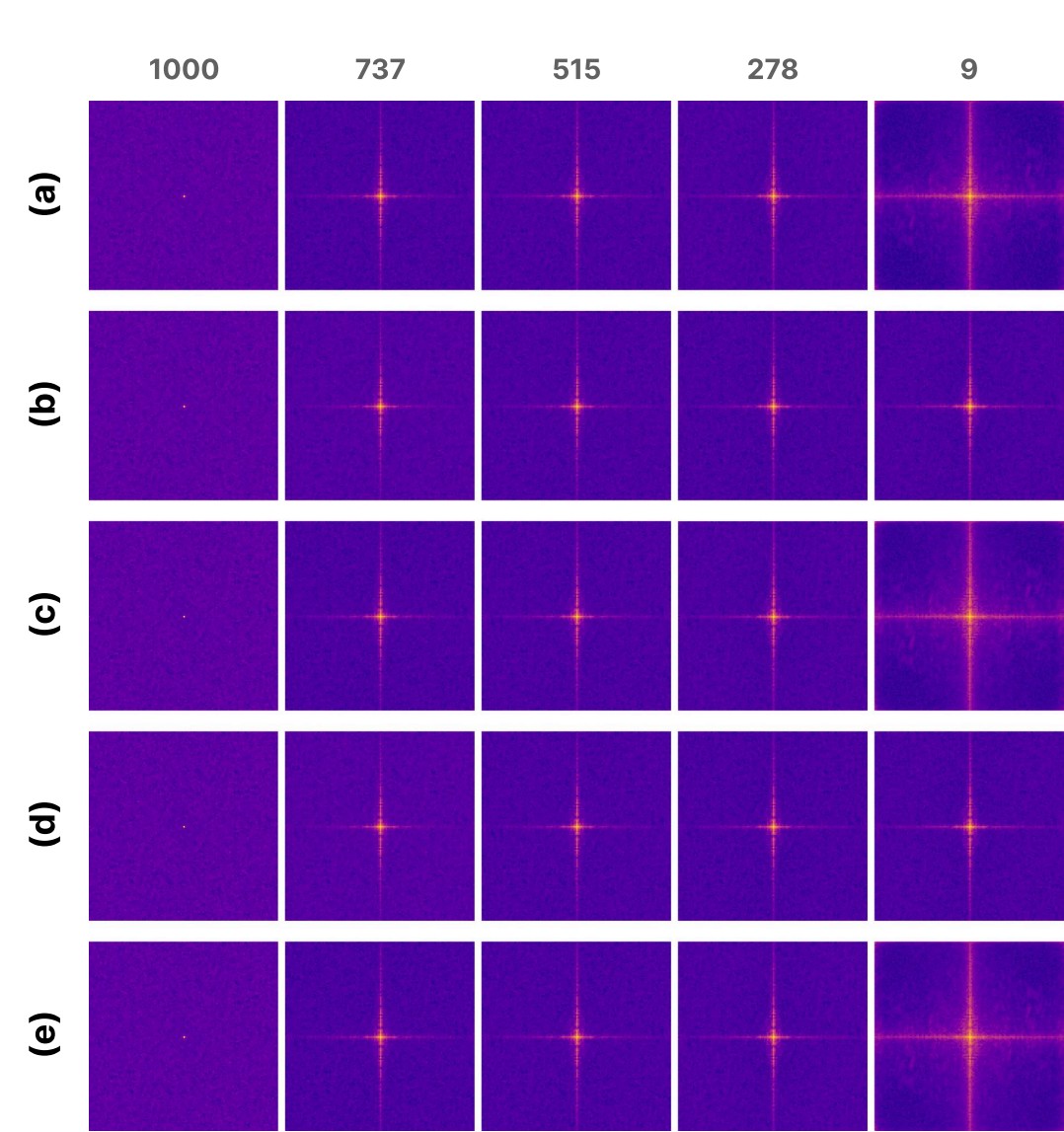

Figure 20: **Frequency Domain Analysis via timestep**. FFT results of each component at timesteps 1000, 737, 515, 278, and 9. (a) Text (b) Text EMA (c) Uncond (d) Uncond EMA (e) Delta3

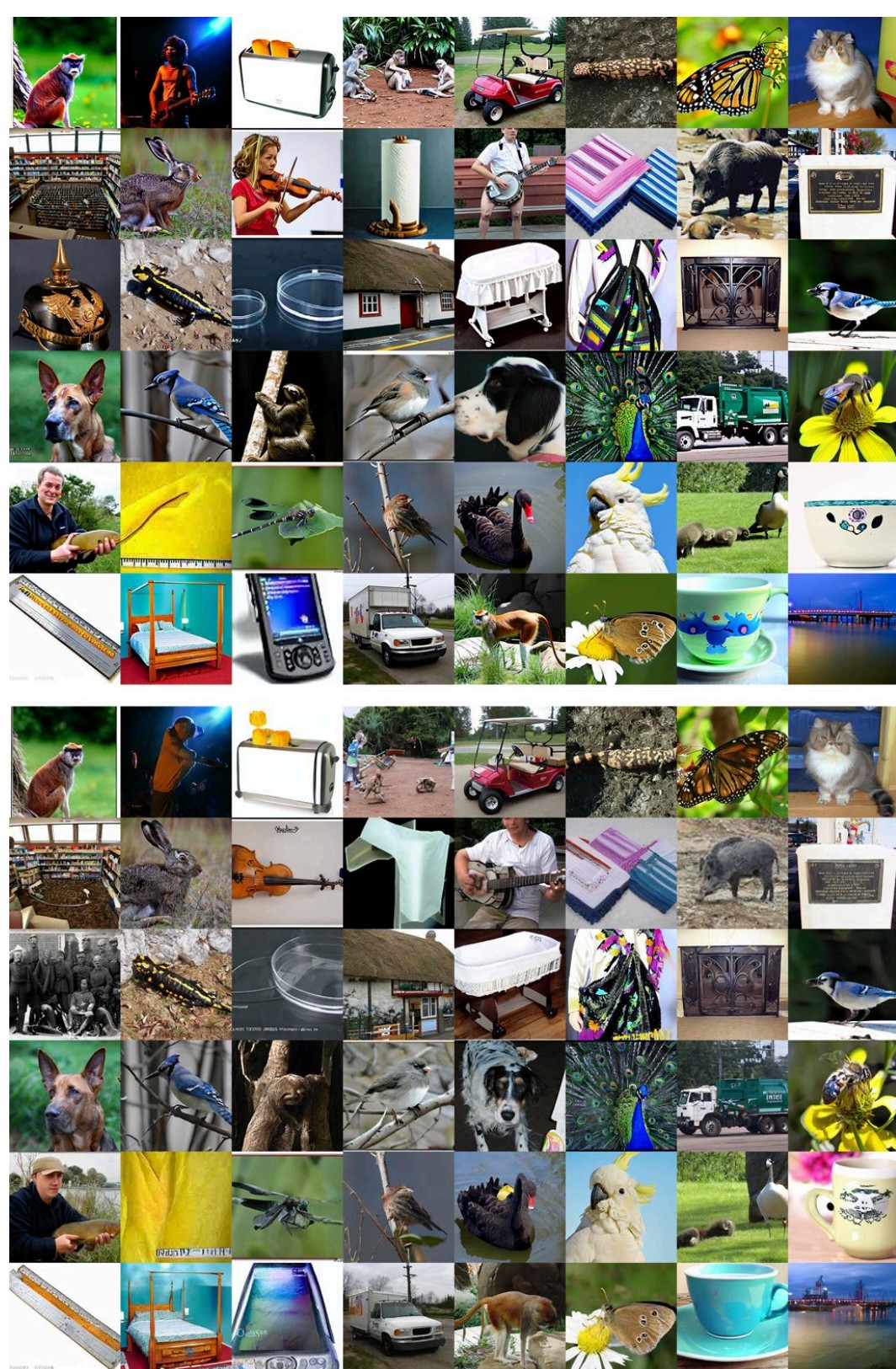

Figure 21: **ImageNet dataset comparison with EDM2-s**. Qualitative comparison with EDM2-s using CFG (top) and REG (bottom).

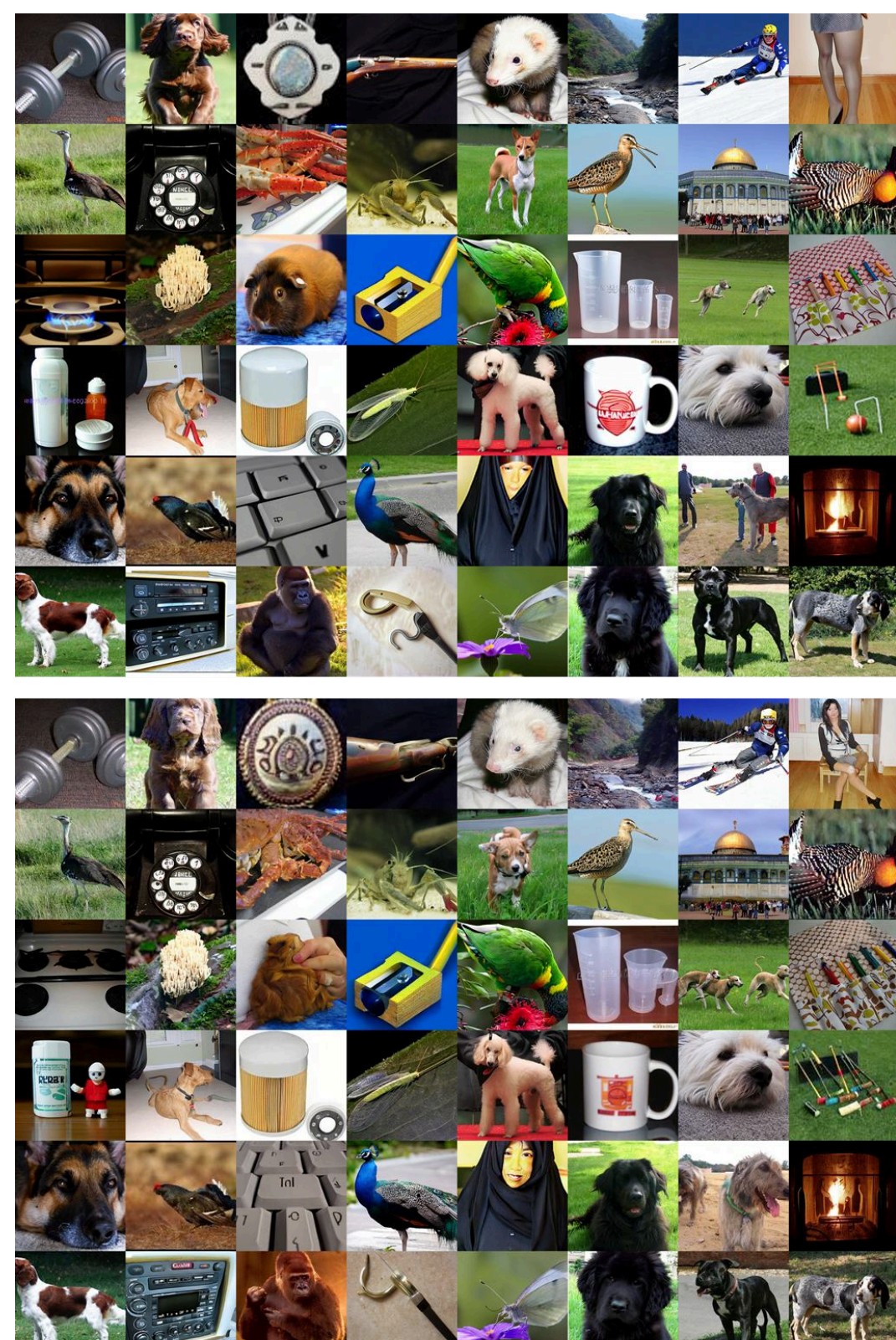

Figure 22: **ImageNet dataset comparison with EDM2-xxl**. Qualitative comparison with EDM2-xxl using CFG (top) and REG (bottom).

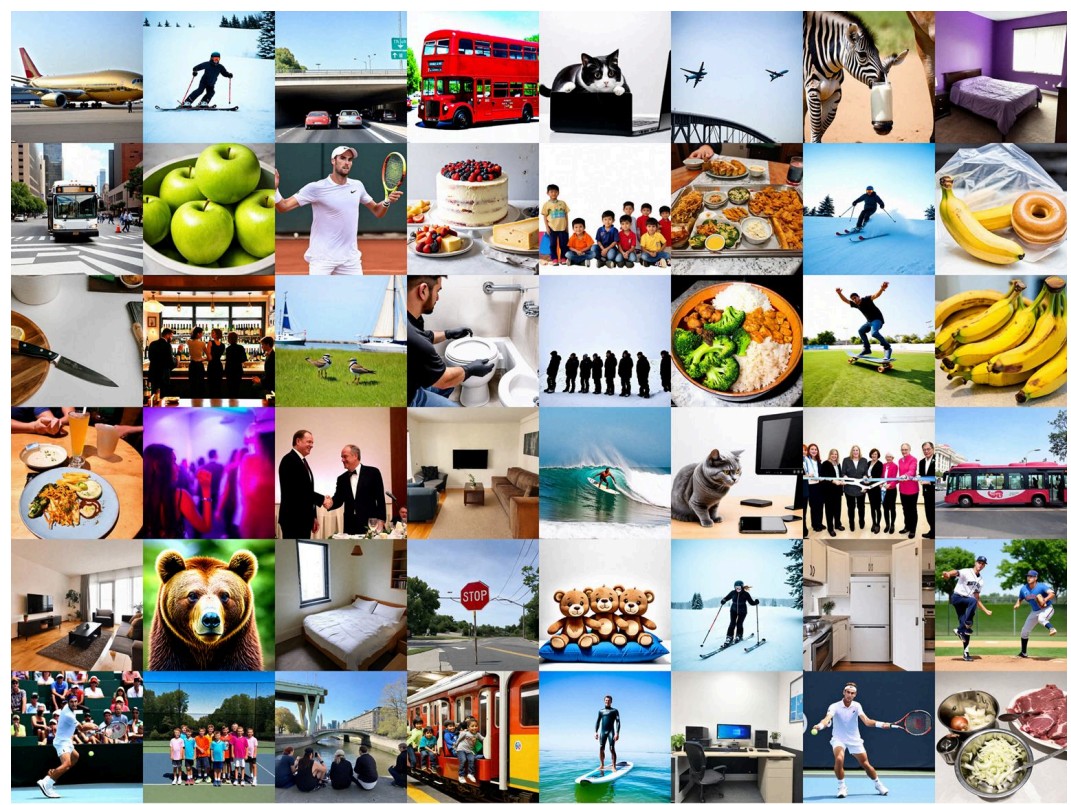

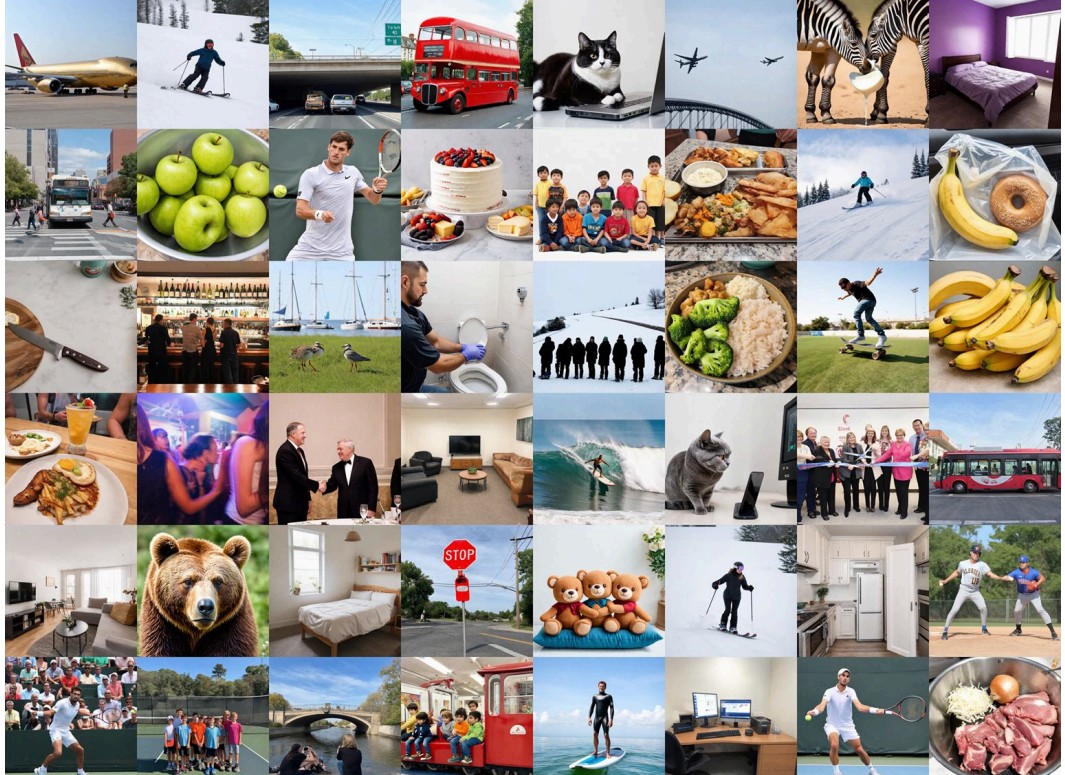

Figure 23: **MS-COCO dataset comparison with SD3**. Qualitative comparison with Stable Diffusion 3 using CFG (top) and REG (bottom).

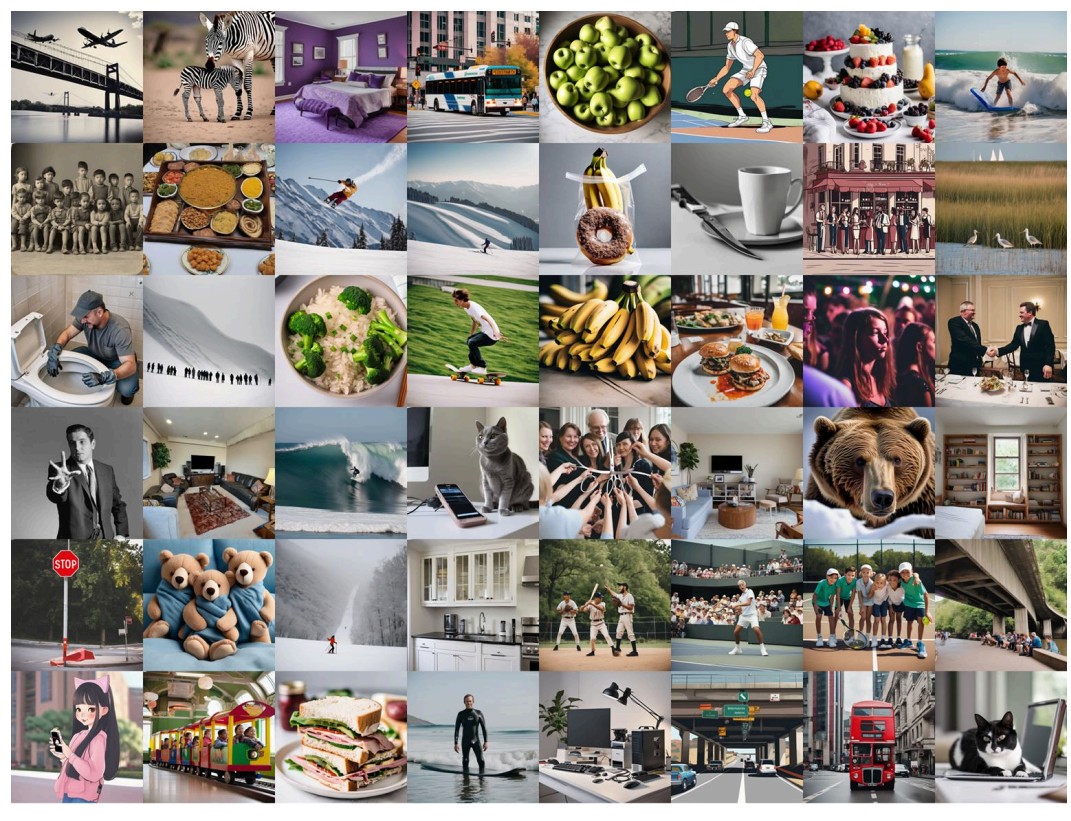

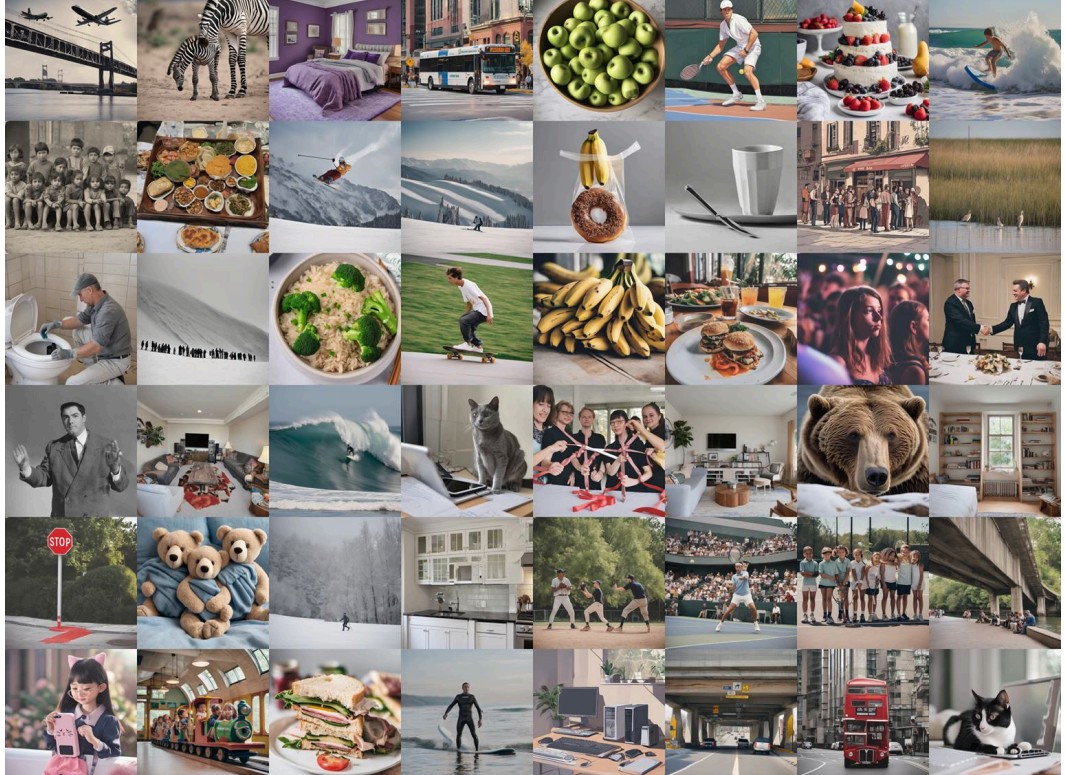

Figure 24: **MS-COCO dataset comparison with SDXL**. Qualitative comparison with Stable Diffusion XL using CFG (top) and REG (bottom).

0.1 ⟶ γ Scale ⟶ 0.9

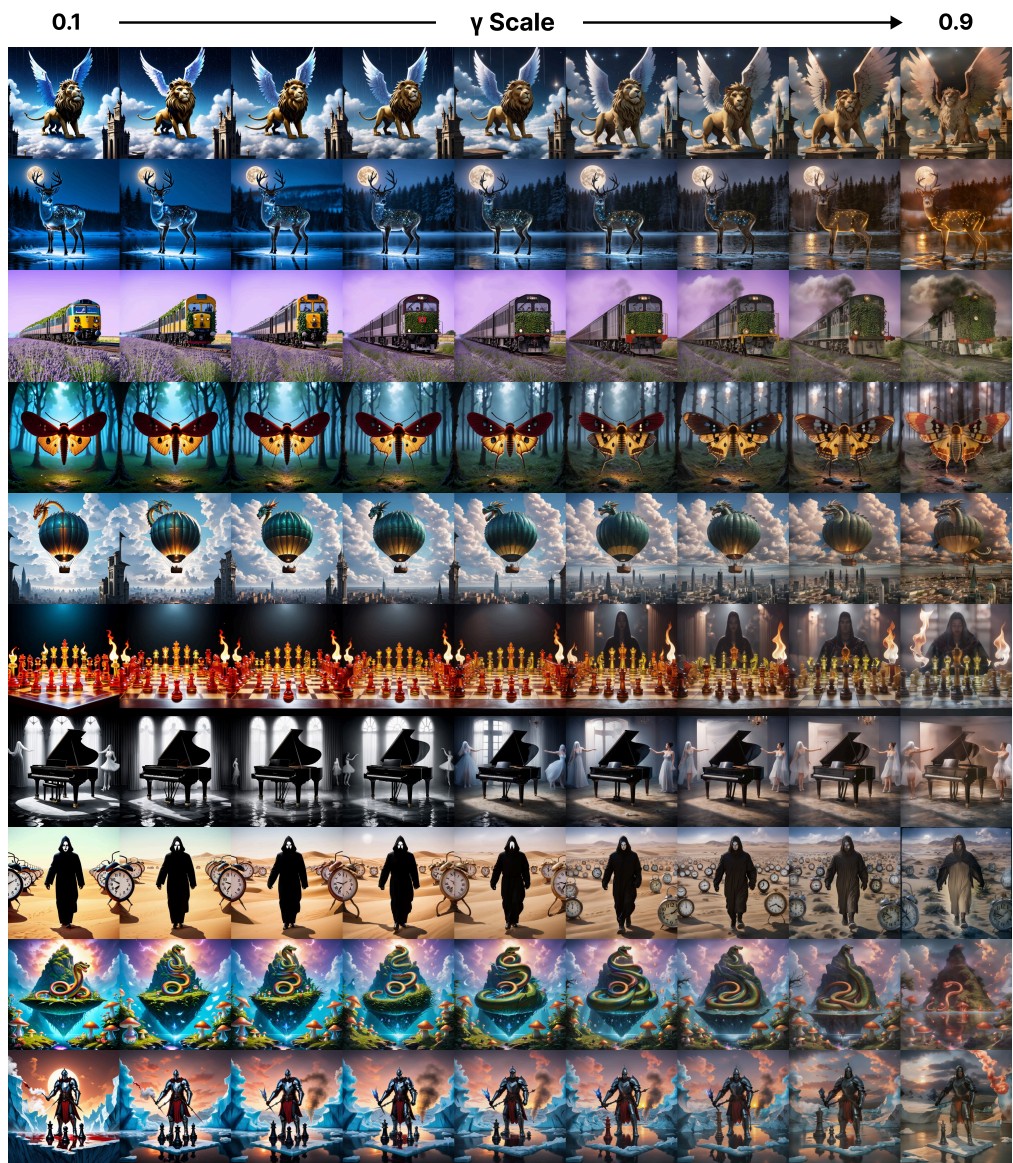

Figure 25: **Additional gamma scale (γ) comparison**. Across gamma scales from 0.1 to 0.9 (step 0.1). Prompts : *"In a city suspended above the clouds, a crystal-winged lion guards the last library of forgotten knowledge as stardust rains gently from the sky.", "A luminous deer with constellation-patterned fur stands motionless on a frozen lake, as time slows to a whisper under the full moon.", "A train made of vines and marble glides through a lavender sky, carrying passengers made of smoke and forgotten lullabies.", "Beneath a forest of glass trees, a moth made of velvet and smoke guides lost travelers with its flickering light.", "A dragon-shaped hot air balloon floats above an inverted city, casting mirrored shadows onto the clouds below.","A chessboard the size of a village hosts a battle between living pieces made of glass and flame, overseen by a silent queen.", "A piano with no keys plays itself in a sunken ballroom, as ghostly dancers sway in rhythm with invisible music.", "A cloaked figure with a mirror for a face walks through a desert of clocks, each ticking in a different language.", "A rainbow serpent coils around a floating mountain, guarding a garden of glass mushrooms and whispered wishes.", "A knight of smoke duels his reflection atop a glacier carved like a chess piece under a bleeding sunset."*

1944
1945
1946
1947
1948
1949
1950
1951
1952
1953
1954
1955
1956
1957
1958
1959
1960
1961
1962
1963
1964
1965
1966
1967
1968
1969
1970
1971
1972
1973
1974
1975
1976
1977
1978
1979
1980
1981
1982
1983
1984
1985
1986
1987
1988
1989
1990
1991
1992
1993
1994
1995
1996
1997

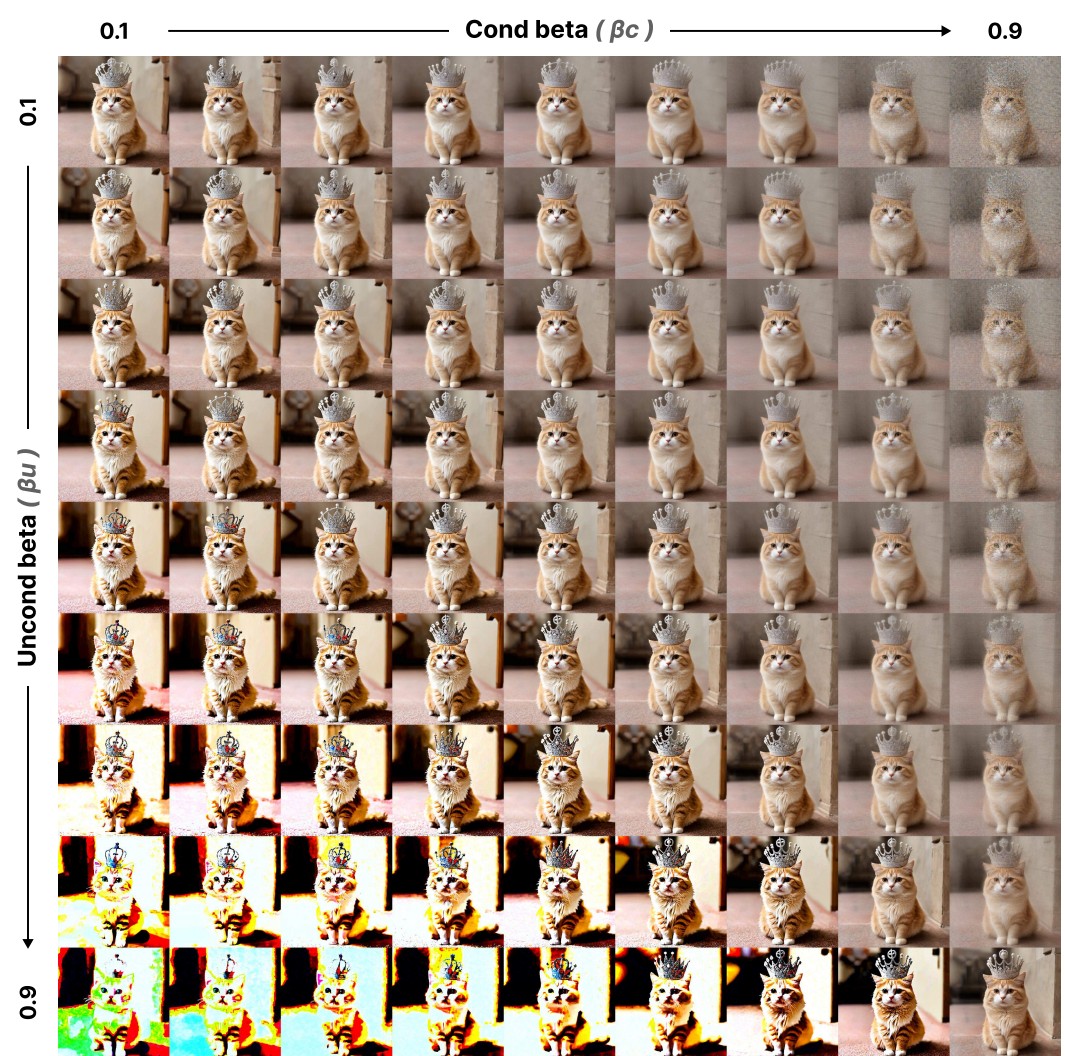

Figure 26: **Additional beta scale ($\beta$) comparison**. Prompt : *"A cat with a queen hat"*

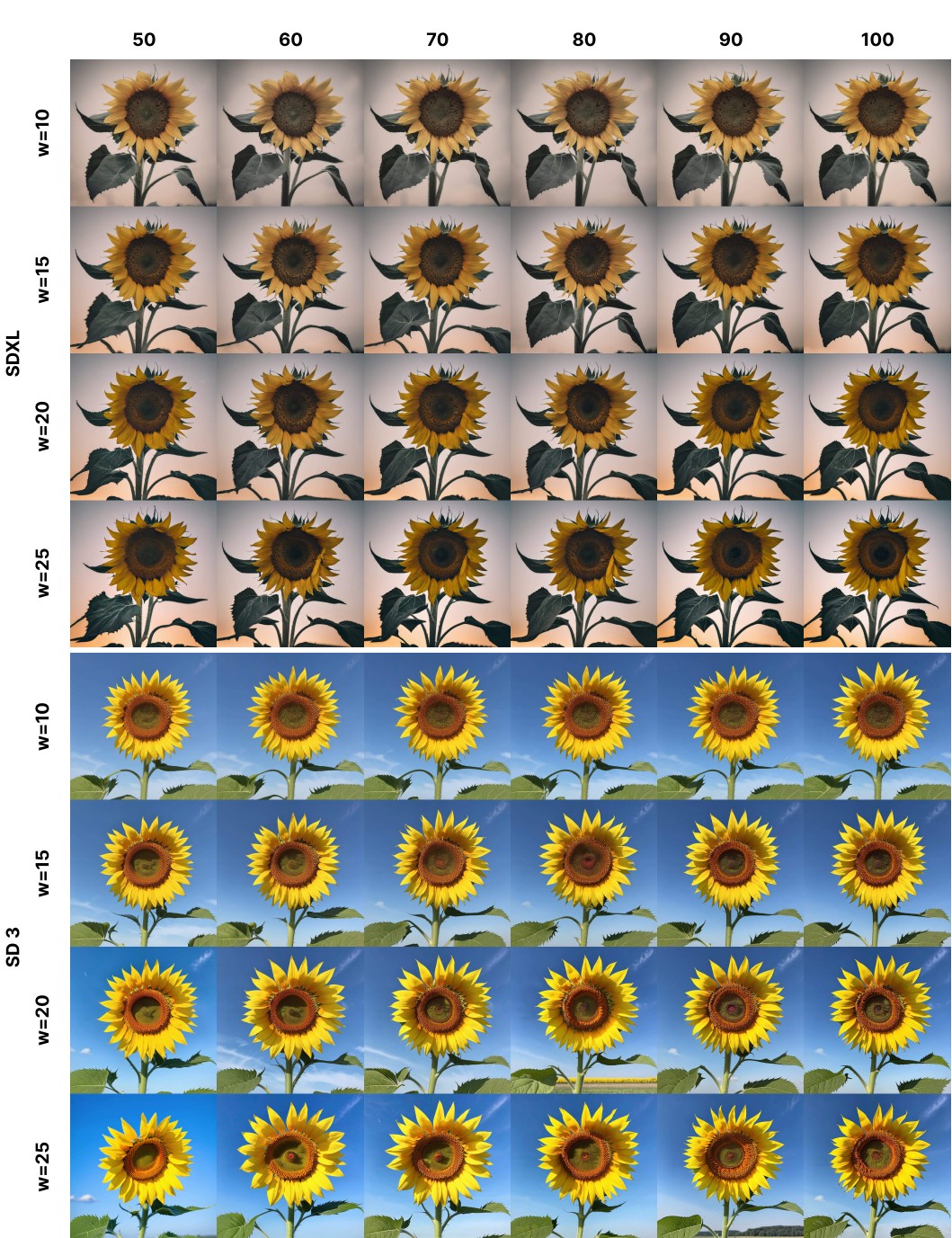

Figure 27: **Various sampling steps.** Images generated at different sampling steps using the best settings for SDXL and SD3. Increasing the timestep does not degrade image quality.

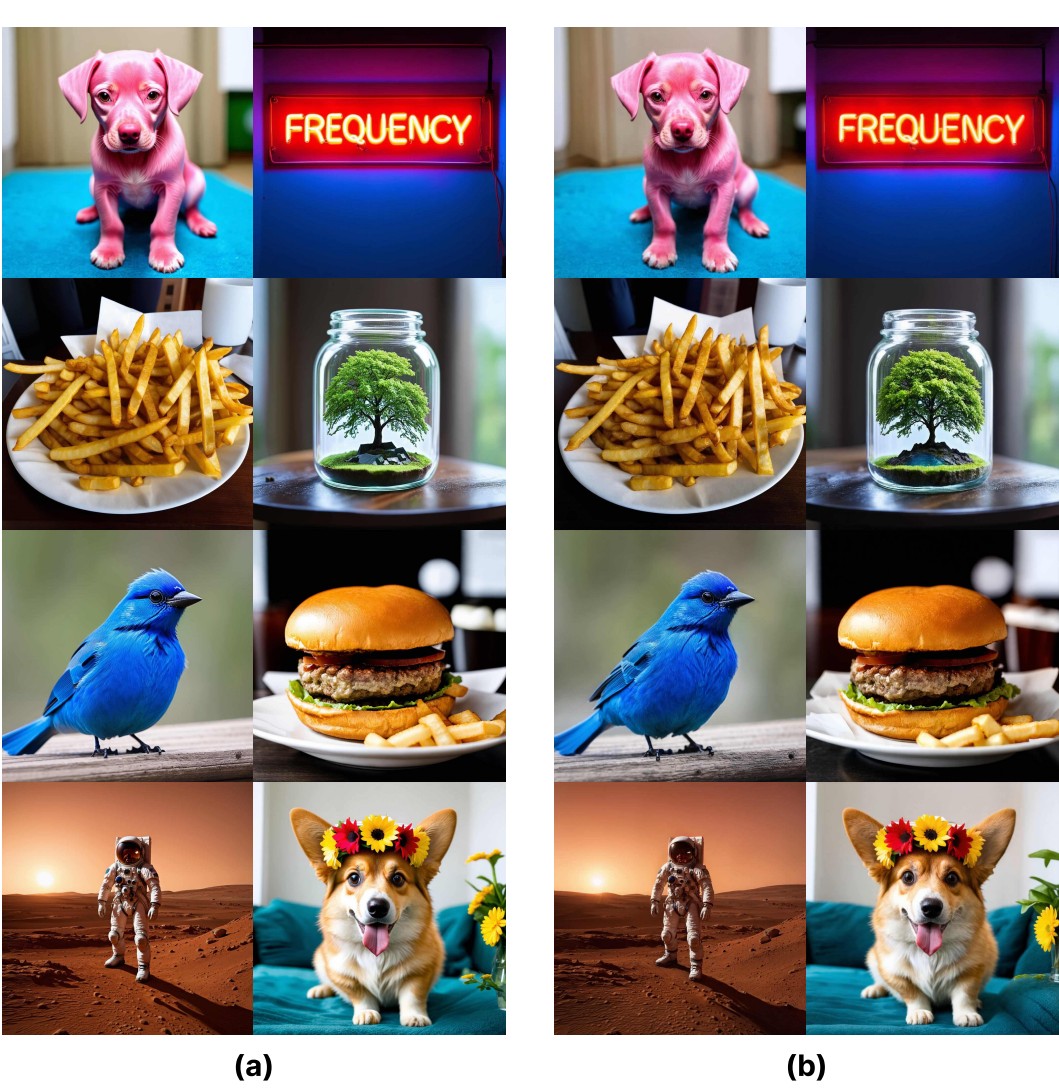

(a)               (b)

Figure 28: **Applying REG in different parameterization in SD3**. Applying REG in both denoised prediction $D$ and velocity $u$ does not make significant differences and demonstrate the robustness of REG in different parameterization. (a) applied REG in $D$, (b) applied REG in $\epsilon$.

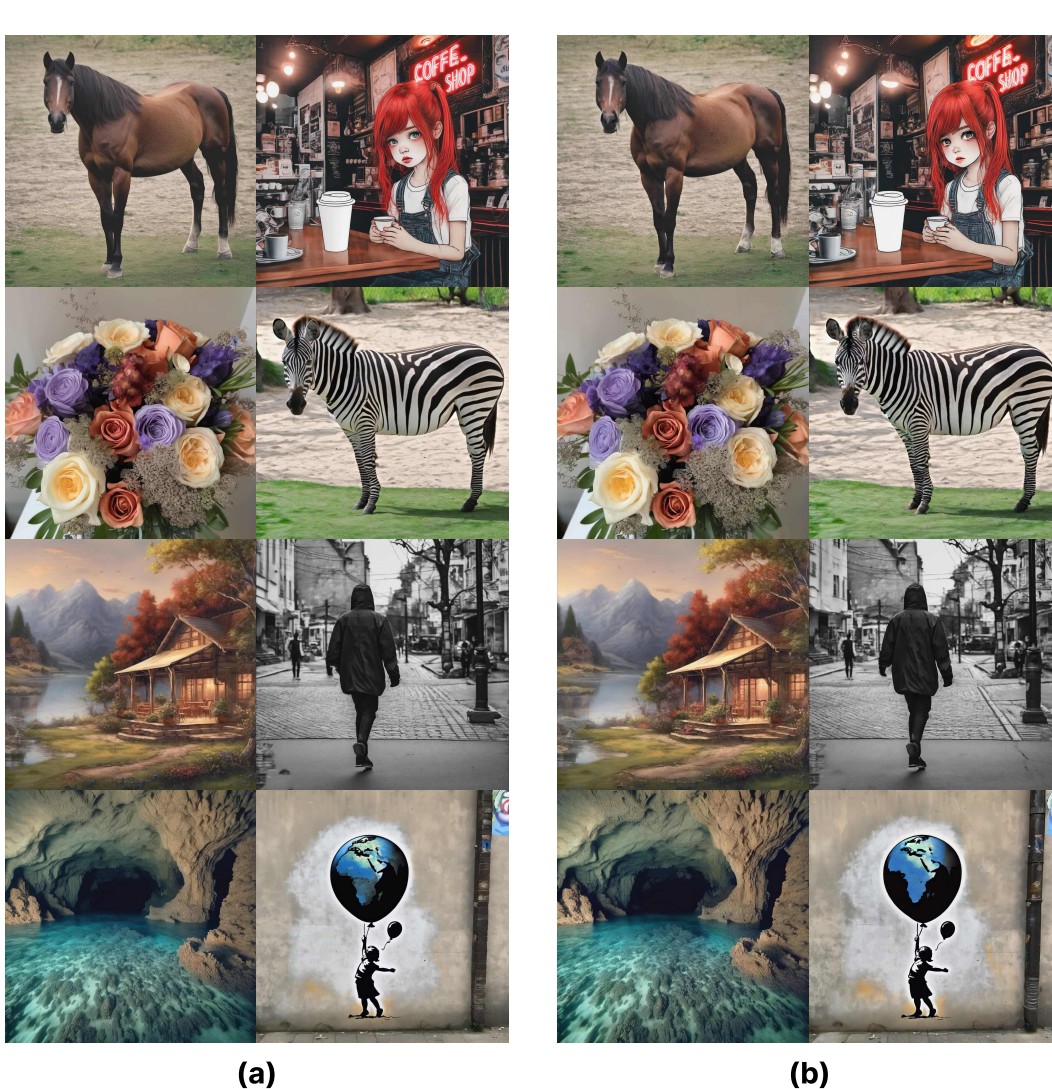

(a)                                                    (b)

Figure 29: **Applying REG in different parameterization in SDXL**. Applying REG in both denoised prediction $D$ and epsilon prediction $\epsilon$ does not make significant differences and demonstrate the robustness of REG in different parameterization. (a) applied REG in $D$, (b) applied REG in $\epsilon$.

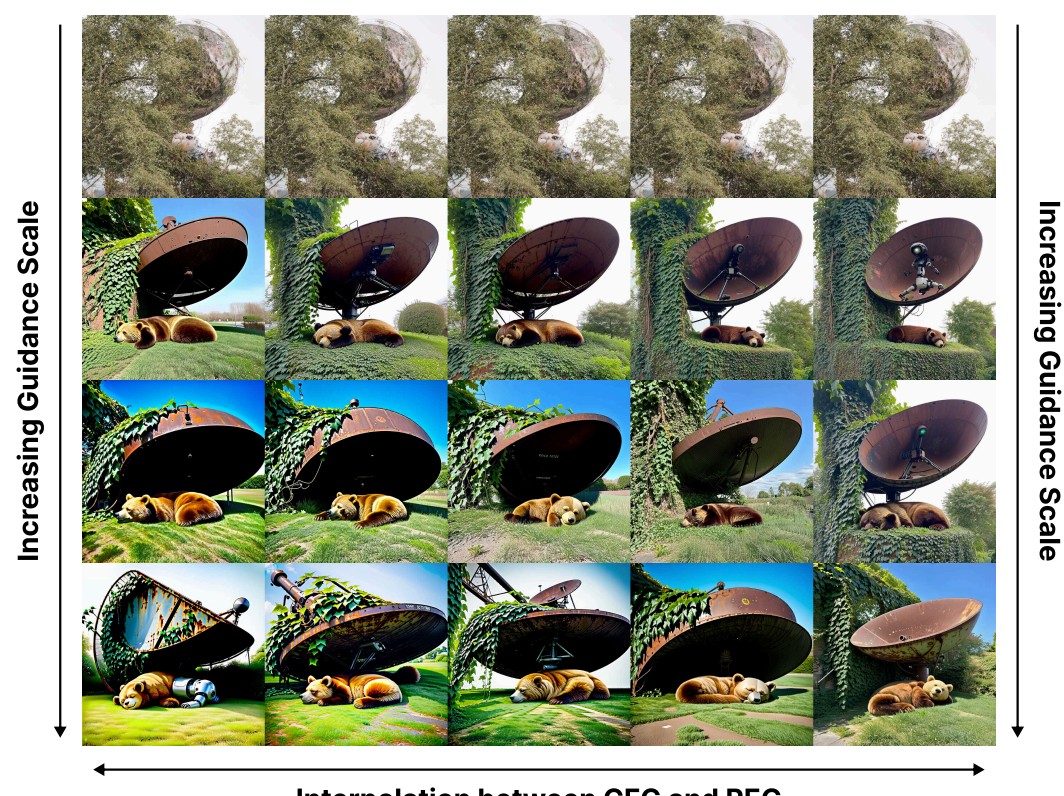

Figure 30: Results for SD3 using the prompt "A bear sleeps under a rusted satellite dish overgrown with ivy". The leftmost column shows results for CFG and the rightmost for REG, while the middle columns correspond to blending between the two.

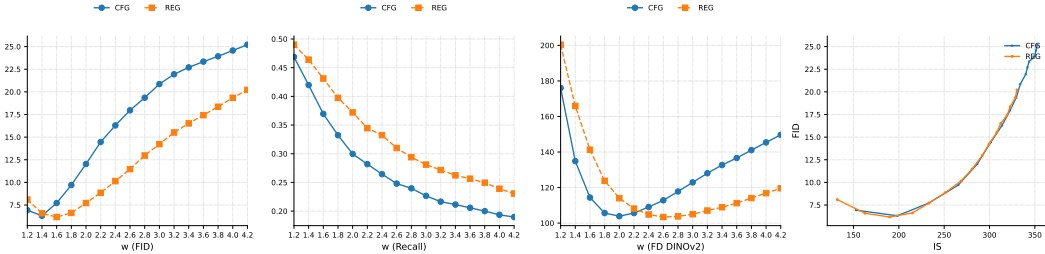

Figure 31: **The diversity-preserving effect of REG on DiT-XL/2.** We evaluate the Recall scores at the points where the quality metrics (FID and FD_DINOv2) reach their respective peaks. REG consistently exhibits higher Recall than CFG in both metrics, demonstrating a superior quality-diversity trade-off.

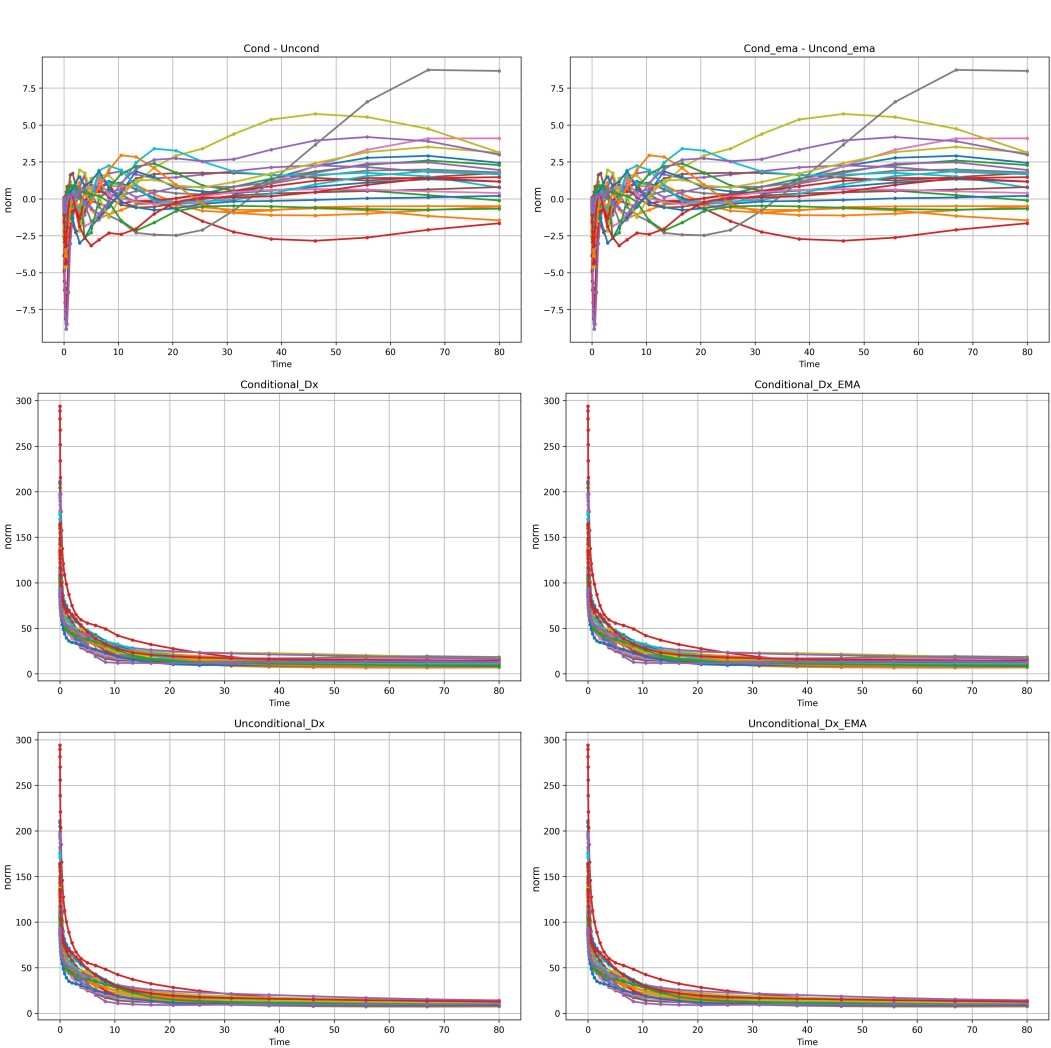

Figure 32: **Comparison of norm stability between vanilla and EMA predictions in EDM2-s.** Using 25 random samples, we tracked the norm evolution for the difference term, which is $\hat{u}(x \mid y) - \hat{u}(x), s^c \hat{u}^m(x \mid y) - s^u u^m(x)$ (top), conditional term, which is $\hat{u}(x \mid y), \hat{u}(x)$ (middle), and unconditional term, which is $\hat{u}(x), \hat{u}^m(x)$ (bottom) components. Both vanilla (left) and EMA (right) configurations show consistent norm magnitudes and stability throughout the 80 sampling steps, indicating that EMA maintains the structural properties of the model predictions.

