# OpenReview forum: "Rectifying Diffusion Guidance with Exponential Moving Average"
_ICLR.cc/2026/Conference — Submitted to ICLR 2026_

### Official Review · Reviewer_uiNj · 2025-10-22

**Soundness:** 1
**Presentation:** 1
**Contribution:** 1
**Rating:** 2
**Confidence:** 5

**Summary:**

This paper proposes a technically novel guidance approach, **Rectified EMA Guidance (REG)**, which replaces the unconditional prediction with a new EMA-rectified prediction. The authors state that REG achieves a better quality–diversity trade-off than widely used CFG. They conduct experiments on both text-to-image and class-conditional tasks to support their claims. REG does not rely on an extra model and shows good compatibility with publicly released models such as SDXL and SD3.

**Strengths:**

The paper proposes REG, which leverages EMA over model predictions during sampling, and provides an analysis of how EMA acts as a low-pass filter in sampling. This is novel and interesting.

**Weaknesses:**

* The current draft is not ready for submission to a venue like ICLR. The manuscript appears hurried and suffers from numerous unprofessional issues, including a line-break problem (line 240), placeholders (line 323, “Figure X”), disordered figure/table numbering (e.g., Figure 4; Tables 1–2 include subfigures/tables that belong to different caption types), and low-resolution, pixelated figures, among others.
* The methodology is not well motivated. I do not find sufficient intuition to justify the current design choices in REG. For example, the authors should explain why they use the EMA difference $\hat{u}^m(x\mid y)-\hat{u}^m(x)$ to rectify the unconditional output rather than using it directly as a guidance signal.
* In practice, the CFG weight $w$ is an effective knob to align predictions with user preferences and to control the quality–diversity trade-off. While REG expands the range over which $w$ can be adjusted, it also introduces several additional hyperparameters to tune $(\beta_1, \beta_2, \gamma)$ and a wider tunable scope for $w$. I do not see clear benefits that justify the increased tuning complexity. The authors argue that REG is a “free lunch” that improves quality without sacrificing diversity, but the current experiments do not support this claim.
* The expansion of REG is $\tilde{u}_t(x\mid y)=\hat{u}_t(x)+w(\hat{u}_t(x\mid y)-\hat{u}_t(x))+\gamma(1-w)(s_1\hat{u}^m_t(x\mid y)-s_2\hat{u}^m_t(x))$. Since $\gamma(1-w)<0$, REG extends the tunable range of $w$. Note that setting $w=2.5,\ \beta_1=0,\ \beta_2=0,\ \gamma=0.5$ (i.e., no EMA in the rectified term) reduces the method to CFG with an effective $w = 1.6$. Referring to Table 2, CFG with $w=1.7$ performs no worse than REG with $w=2.5$ (CFG with $w=1.6$ is not reported, but it is reasonable to expect it would be at least comparable to $w=1.7$). This suggests the EMA component may even harm generation quality.
* The motivation to replace AG based on the availability of “weak” models feels forced. Although early releases such as SDXL and SD3 may not include a weak model, such models can be saved during training. There is no obvious obstacle preventing future releases from supporting AG if maintainers choose to do so. Notably, in EDM-S, REG performs only comparably to vanilla CFG and still lags far behind AG.

**Questions:**

* The authors should provide qualitative comparisons following [1], or quantitative comparisons such as precision–recall or an FID–IS diagram, following Figure 5 in [2], to support their claim.
* EMA behavior is sensitive to the cumulative number of sampling steps; the authors should analyze how the number of steps influences the generation behavior of REG.
* The authors should provide vector graphics or higher-resolution figures for Figure 2, Figure 3, Table 1, and Table 2.
* The colorful notations appear unnecessary. Unless they convey essential information, please avoid them.

[1] Guiding a Diffusion Model with a Bad Version of Itself

[2] Classifier-Free Diffusion Guidance

---

> ### Author Response · Authors · 2025-11-23
>
> We sincerely thank you for your time and effort in reviewing our paper. We appreciate your positive comments on:
>
> - the novelty of REG and the analysis of EMA as a low-pass filter
>
> Please find our responses to your comments below.
>
> ---
>
> **C1. Concerns regarding Presentation**
>
> > - **W.1** The current ...
> > - **Q.3** The authors ...
> > - **Q.4** The colorful ...
>
> We sincerely appreciate your constructive feedback. We apologize for the confusion caused by the presentation issues. Regarding the comment on Line 240, we clarify that the sentence appeared incomplete due to a layout artifact where the text flow was unintentionally interrupted by Figure 3. We have adjusted the layout to ensure a seamless narrative flow. In the revised manuscript, we have refined the writing throughout and improved the overall presentation to enhance clarity and readability. We have also enhanced the quality of the image visualizations as suggested. Additionally, in response to your feedback, we have removed all color notation from the equations.
>
>
> **C2. Motivation and Justification of Design Choices**
>
> > - **W.2** The methodology ...
>
> We explicitly discussed the motivation behind our design choices, particularly the reason for not using the simple EMA difference directly, in Section 3.1 (Lines 86-87) and Section 4 (Lines 235-236) of the original submission. To clearly address the concern raised by your example, we reiterate the intuition provided in the manuscript: The simple EMA difference term $\hat u^m(x \mid y) - \hat u^m(x)$ cannot be used directly because its frequency level differs from the conditional term. Specifically, this difference term is dominated by low-frequency conditional components (Figure 4-(b)). Therefore, simply adding or subtracting it leads to spectral inconsistencies. To resolve this, we add the difference term to the unconditional prediction to synthesize a 'pseudo-weak conditional prediction' that aligns with the frequency levels of the target. This design effectively emulates the weak conditional mechanism found in AG. By utilizing this term to subtract the conditional bias from the CFG boosting term, we successfully mitigate oversimplification and enhance generation quality while preserving sample diversity.
>
>
> **C3. Concerns regarding Hyperparameter Complexity**
>
> > - **W.3** In practice, ...
>
> To simplify the parameter complexity, we have unified $\beta_c$ and $\beta_u$ into a single parameter $\beta$, and fixed to 0.9. Experiments exploring different $\beta_c$ and $\beta_u$ values are provided in Appendix I (now reported in Figures 10 and 26) for reference. Regarding $\gamma$, we designed it to depend on $w$ (specifically, scaling linearly with $w$) to minimize the search space. In our experiments, a coefficient of approximately 0.4 was effective for most models. Consequently, $w$ remains the sole parameter requiring tuning, typically set to values higher than standard CFG scales.
>
> In response to the concern about added tuning complexity, we emphasize that REG does not introduce meaningful extra hyperparameter search in practice. As shown in the $\beta_c/\beta_u$ ablations (Figs. 10 and 26), when $\beta_c>\beta_u$ the rectified term becomes overly smoothed, leading to blurry outputs, whereas $\beta_u>\beta_c$ amplifies low-frequency conditional bias and results in oversaturation (noted around Lines 1231–1236). Based on these observations, and consistent with common EMA practice in the diffusion literature, we set $\beta_c=\beta_u=\beta=0.9$; our local sweeps around $\beta=0.9$ show no visually significant differences, indicating that $\beta$ is not a sensitive tuning knob. Likewise, the $\gamma$ ablation (Fig. 25) shows that $\gamma\approx 0.4$ yields the best visual quality across models. Therefore, in the intended operating regime of REG, $\beta$ and $\gamma$ are fixed by design, leaving the guidance scale $w$ as the only parameter that users tune—exactly as in standard CFG—while REG simply expands the stable and effective range of $w$.

---

> ### Author Response · Authors · 2025-11-23
>
> **C4. Comparison with CFG and Effectiveness of proposed Rectified Component**
>
> > - **W.4** The expansion ...
>
>
> We respectfully point out that the reviewer's assumption relies on a configuration ($\beta=0$) that is explicitly excluded from our framework. As defined in Line 202, our method strictly constrains $\beta$ to the range $(0, 1)$, with $\beta=0.9$ used as the standard setting in our experiments. Assuming $\beta=0$ implies removing the EMA component entirely, which naturally reverts the method to a basic form; this is analogous to stating that CFG becomes identical to unconditional sampling when the guidance scale is 0. This assumption contradicts the intended mechanism of REG.
>
> Regarding the observations on Table 2 (now reported in Figure 8), we have expanded the guidance scale analysis in the table below to demonstrate that REG is not merely a "widened" version of CFG. These results demonstrate that REG maintains Recall levels comparable to CFG, highlighting its distinct capability beyond a simple extension. First, at the same guidance scale of $w=2.5$, REG significantly outperforms CFG in both quality and diversity (FID: 6.37 vs. 10.11; Recall: 0.70 vs. 0.62). While CFG suffers from severe oversimplification and quality degradation at this scale, REG maintains structural stability. Even when comparing points of similar diversity (Recall $\approx$ 0.69), REG ($w=2.7$) achieves a superior FID (6.73) compared to CFG ($w=1.7$, FID 7.13). This confirms that REG provides a better trade-off. This confirms the existence of distinct operating points where REG offers advantages. Consistently, Figs. 31 show guidance-scale sweeps on DiT-XL/2 and EDM-s. Across the full sweep, REG yields a consistently better FID–Recall trade-off than CFG: for comparable (or lower) FID, REG maintains higher Recall, and Recall degrades more slowly as w increases. Also, it is important to note that REG is explicitly designed to counteract the over-accumulation of conditional components at high guidance levels. Consequently, REG's optimal operating point (peak) naturally shifts to a higher guidance scale compared to CFG. The original sweep in Table 6 did not sufficiently cover this higher range to capture REG's true peak. To address this, we have expanded the analysis with finer granularity and extended the range to higher guidance scales, as detailed in the updated table below. These results clearly demonstrate that at its true peak, REG outperforms CFG in both image quality (FID) and diversity (Recall).
>
> Consequently, we strongly refute the conjecture that the EMA component harms generation quality. On the contrary, the EMA component plays a crucial role in high-guidance regimes by disentangling the conditional and quality components within the CFG boosting term. Specifically, it effectively subtracts the conditional bias, thereby mitigating the oversaturation and oversimplification issues inherent to CFG. This cancellation effect is empirically demonstrated in Figure 7 (fractal-like Gaussian mixture), where REG restores the unconditional distribution and captures the full mode structure—effectively resolving the concentration issues inherent to CFG. We further validate this distinction through visualized trend plots (Figures 5 and 8), experiments on the DiT-XL model (Figure 31) and extensive qualitative results (Figures 1, 16–19) that visually confirm improvements in both perceptual quality and diversity.

---

> ### Author Response · Authors · 2025-11-23
>
> Table: Extended Results on EDM2-XXL
>
> | CFG | 1.1 | 1.3 | 1.5 | 1.7 | 1.9 | 2.1 | 2.3 | 2.5 | 2.7 | 2.9 | 3.1 |
> | :--- | :--- | :--- | :--- | :--- | :--- | :--- | :--- | :--- | :--- | :--- | :--- |
> | **FID** | 2.4 | 4.01 | 5.72 | 7.13 | 8.21 | 9.03 | 9.62 | 10.11 | - | - | - |
> | **FDDINOv2** | 45 | 36.72 | 38.02 | 42.93 | 48.89 | 54.6 | 60.29 | 65.76 | - | - | - |
> | **Precision** | 0.61 | 0.62 | 0.61 | 0.59 | 0.58 | 0.56 | 0.55 | 0.53 | - | - | - |
> | **Recall** | 0.74 | 0.73 | 0.71 | 0.69 | 0.67 | 0.66 | 0.64 | 0.62 | - | - | - |
>
> | REG | 1.1 | 1.3 | 1.5 | 1.7 | 1.9 | 2.1 | 2.3 | 2.5 | 2.7 | 2.9 | 3.1 |
> | :--- | :--- | :--- | :--- | :--- | :--- | :--- | :--- | :--- | :--- | :--- | :--- |
> | **FID** | 2.13 | 2.67 | 3.42 | 4.14 | 4.8 | 5.44 | 5.96 | 6.37 | 6.73 | 7.01 | 7.25 |
> | **FDDINOv2** | 50.17 | 41.45 | 37.89 | 36.93 | 37.54 | 38.94 | 40.87 | 43.12 | 45.23 | 47.50 | 49.73 |
> | **Precision** | 0.61 | 0.62 | 0.62 | 0.62 | 0.62 | 0.61 | 0.61 | 0.60 | 0.60 | 0.59 | 0.59 |
> | **Recall** | 0.74 | 0.74 | 0.73 | 0.72 | 0.72 | 0.71 | 0.71 | 0.70 | 0.69 | 0.69 | 0.68 |
>
> Table: Extended EDM2-S Table
> | CFG | 1.2 | 1.4 | 1.6 | 1.8 | 2 | 2.2 | 2.4 | 2.6 | 2.8 | 3 | 3.2 | 3.4 | 3.6 | 3.8 | 4 | 4.2 |
> | ---------- | ------ | ------ | ------ | ------ | ------ | ------ | ------ | ------ | ------ | ------ | ------ | ------ | ------ | ------ | ------ | ------ |
> | **FID** | 4.04 | 3.45 | 4.82 | 6.89 | 9.14 | 11.34 | 13.38 | 14.98 | 16.53 | 17.73 | 18.78 | 19.73 | 20.47 | 21.15 | 21.70 | 22.15 |
> | **FD_DINOv2** | 143.18 | 102.36 | 81.25 | 72.87 | 70.98 | 72.61 | 76.12 | 80.62 | 85.48 | 90.63 | 95.52 | 100.49 | 105.04 | 109.43 | 113.45 | 117.50 |
> | **Precision** | 0.44 | 0.54 | 0.61 | 0.67 | 0.70 | 0.73 | 0.75 | 0.76 | 0.77 | 0.78 | 0.78 | 0.78 | 0.78 | 0.78 | 0.78 | 0.78 |
> | **Recall** | 0.28 | 0.24 | 0.21 | 0.19 | 0.17 | 0.16 | 0.14 | 0.13 | 0.12 | 0.11 | 0.11 | 0.10 | 0.09 | 0.09 | 0.09 | 0.08 |
>
> | REG | 1.2 | 1.4 | 1.6 | 1.8 | 2 | 2.2 | 2.4 | 2.6 | 2.8 | 3 | 3.2 | 3.4 | 3.6 | 3.8 | 4 | 4.2 |
> | ---------- | ------ | ------ | ------ | ------ | ------ | ------ | ------ | ------ | ------ | ------ | ------ | ------ | ------ | ------ | ------ | ------ |
> | **FID** | 5.17 | 3.70 | 3.41 | 3.87 | 4.80 | 6.02 | 7.37 | 8.74 | 10.03 | 11.26 | 12.47 | 13.52 | 14.48 | 15.48 | 16.43 | 17.12 |
> | **FD_DINOv2** | 166.69 | 131.80 | 108.06 | 91.81 | 81.25 | 75.13 | 72.06 | 70.71 | 70.81 | 72.01 | 73.87 | 76.12 | 78.93 | 81.53 | 84.67 | 87.80 |
> | **Precision** | 0.42 | 0.48 | 0.53 | 0.58 | 0.62 | 0.66 | 0.68 | 0.71 | 0.72 | 0.74 | 0.75 | 0.76 | 0.77 | 0.78 | 0.78 | 0.78 |
> | **Recall** | 0.31 | 0.27 | 0.25 | 0.23 | 0.22 | 0.21 | 0.19 | 0.18 | 0.17 | 0.16 | 0.15 | 0.15 | 0.14 | 0.14 | 0.14 | 0.13 |
>
> DiT-XL peak performance
>
> | | FID | Recall |
> | :--- | --- |  ---: |
> | CFG ($w=1.4$) | 3.45 | 0.24 |
> | REG ($w=1.6$) | **3.41** | **0.25** |

---

> ### Author Response · Authors · 2025-11-23
>
> **C5. Justification for Training-free Approach against Auto-Guidance**
>
> > - **W.5** The motivation ...
>
> While we acknowledge the theoretical merits of Auto-Guidance (AG), applying it to large-scale models like Stable Diffusion presents significant practical limitations:
>
> 1. Computational & Memory Overhead: Even if pre-trained weak conditional checkpoints were available, utilizing AG requires loading a secondary model alongside the main model. This effectively doubles both GPU memory usage (VRAM) and inference latency. Given the substantial size of modern diffusion models, such overhead renders AG computationally prohibitive for many real-world applications.
>
> 2. Training Complexity: Unlike standard CFG, which relies on the widely established 'label dropping' strategy to train conditional and unconditional distributions within a single model, there is no established paradigm for jointly training 'conditional' and 'weak-conditional' distributions. Developing and verifying such a methodology would require proposing a novel training framework, which is a significantly heavier burden than the minimal hyperparameter tuning required for REG.
>
> Therefore, considering the heavy computational cost and the lack of a standard training protocol for AG, we argue that REG provides a much more practical and efficient solution for diversity preservation.
>
> Furthermore, our work aligns with recent guidance studies including APG, FDG, PAG, SAG, SEG, AG, that also aim to solve the entanglement of conditional fidelity, quality, and diversity in a training-free manner.
>
>
> **C6. Additional Qualitative and Quantitative Comparison**
>
> > - **Q.1** The authors ...
>
> Thank you for the valuable suggestion. We have incorporated both the quantitative FID-IS analysis and additional qualitative comparisons.Regarding the FID-IS curve (now presented in Figure 18), while REG exhibits a U-shaped trajectory similar to CFG, the curve shifts notably further towards the bottom-right. This indicates a superior trade-off, demonstrating that REG generates higher-quality samples (lower FID) at equivalent diversity levels (higher IS). Qualitatively (Figure 30), we compare the results of CFG (2nd row) and REG (4th row). As shown in the examples (e.g., '...a rusted satellite dish...' and 'A bear...'), REG achieves better semantic alignment with the prompt and superior object fidelity compared to CFG. These visual results further corroborate that REG effectively enhances generation quality while preserving the intended diversity.
>
>
> **C7. Analysis on Sensitivity to Sampling Steps**
>
> > - **Q.2** EMA behavior ...
>
> To analyze the sensitivity to sampling steps, we conducted additional experiments visualized in Figure 27. We evaluated both SD3 and SDXL across a wide range of sampling steps (from 50 to 100, in increments of 10) and high guidance scales ($w \in [10, 25]$). The results confirm that the EMA mechanism remains numerically stable and does not diverge as sampling steps increase. Crucially, we observed that the method stably maintains sample diversity throughout the process, demonstrating the robustness of our approach against step accumulation.

---

### Official Review · Reviewer_gP7p · 2025-10-30

**Soundness:** 3
**Presentation:** 3
**Contribution:** 3
**Rating:** 6
**Confidence:** 5

**Summary:**

This paper introduces Rectified EMA Guidance (REG), a simple yet effective modification of classifier-free guidance (CFG) that mitigates the harmful effects of high guidance scales. Inspired by autoguidance and prior guidance techniques, REG employs an exponential moving average (EMA) of the conditional and unconditional model predictions to refine the guidance term in diffusion models. The authors show that REG defines a better negative signal for CFG by leveraging an average of past conditional and unconditional predictions, and—unlike autoguidance—REG requires no additional model training. Experiments demonstrate the advantages of REG over CFG when applied to recent conditional diffusion models.

**Strengths:**

- The method addresses an important aspect of current generative models — the diversity-quality tradeoff. Since CFG is widely used across diffusion models, improving this aspect could have a significant impact on many applications.

- The method is simple to implement and can be integrated into existing models with minimal overhead.

- The approach is well-justified through a frequency analysis of the diffusion process and the filtering behavior of the EMA operator. Moreover, the use of EMA to improve the quality of diffusion model sampling is an interesting research direction.

- The paper is well-written and easy to follow for the most part.

**Weaknesses:**

* With a slight reformulation, this method could be viewed as a generalization of the momentum operation used in APG. I recommend that the authors include this discussion in Section 3.2 to better highlight the differences between APG and REG.

* Some of the quantitative results are somewhat confusing. For instance, IG and CFG perform very similarly in the EDM2 experiment in Table 3, which is usually not the case. After inspecting the hyperparameters in Table 4, it appears that some settings are not fully aligned with best practices for diffusion models. For example, setting $t_{\text{min}} = 2$ for EDM2 with IG might be too restrictive, as it disables guidance during the middle stages of sampling. The paper uses $t_{\text{min}} = 0.28$. Similarly, $w_{\text{high}}$ for FDG is set to 3 for SDXL, whereas the original paper typically uses larger values such as 7 or 10. For a more robust and fair comparison, it would be helpful to adopt a more standardized setup—comparing the proposed method and several baselines across a range of guidance scales and with more practical hyperparameters.

* The images are heavily downsampled and degraded in quality, making it difficult to compare fine details across different outputs given the current image resolution in the paper.

**Questions:**

1. Have you encountered any instability issues when computing the norm, particularly toward the end of the sampling process?

2. Do you have an explanation for why the best CFG result outperforms REG in Table 6?

---

> ### Author Response · Authors · 2025-11-23
>
> We sincerely thank you for your time and effort in reviewing our paper. We appreciate your positive comments on:
>
> - the well-justified approach addressing the diversity-quality trade-off via frequency analysis
> - the training-free and plug-and-play manner
> - the well-written and easy-to-follow presentation
>
> Please find our responses to your comments below.
>
> ---
>
> **C1. Relationship with APG**
>
> > - **W.1** With a slight ...
>
> Thank you for the valuable suggestion. We have addressed this by explicitly clarifying the difference from APG in Section 3.2 (lines 289-290) and providing a comprehensive derivation in Appendix H. We have formulated APG and REG similarly to the CFG formulation to interpret each term and have demonstrated that the roles of the EMA terms are different. We reformulate the APG update rule to reveal its underlying relationship with standard CFG. The decomposition is as follows:
>
> $$
> \begin{aligned}
> \tilde u _{t _k}^{apg}(x \mid y) &= \hat u _{t _k}(x \mid y) + \underbrace{ws _{t _k}(\hat u _{t _k}(x \mid y) - \hat u _{t _k}(x))} _{\text{Down-scaled CFG boosting term}} + \underbrace{w\beta \left(\sum^{k-1} _{i=0}\beta^{i}(s _{t _{k-1-i}}(\hat u _{t _{k-1-i}}(x \mid y)) - \sum^{k-1} _{i=0}\beta^{i}(s _{t _{k-1-i}}(\hat u _{t _{k-1-i}}(x)))\right)} _{\text{Reverse EMA conditional term}} - \underbrace{w(1-\eta) \hat u _{t _k}^\parallel(x \mid y)} _{\text{Parallel conditional term}}
> \end{aligned}
> $$
>
> The APG update rule is driven by three distinct components. The Down-scaled CFG boosting term acts as the primary guidance force, containing both conditional and quality components, but it is uniformly down-scaled by the factor $s_k$. The Reverse EMA conditional term modulates guidance using historical statistics; however, due to the negative decay rate ($\beta < 0$), this term self-cancels (oscillatory nature) at every step. Instead of smoothing, it functions as a "reverse" mechanism that amplifies local variation of the model prediction, emphasizing immediate changes rather than long-term trends. Finally, the Parallel conditional term attempts to disentangle the conditional and quality components by geometrically suppressing the projection of the combined term onto the direction of the conditional prediction.
>
> While APG attempts to decouple the conditional and quality components via geometric projection, REG approaches this separation by leveraging the Low-Pass Filter (LPF) characteristics of EMA. The REG update rule is derived as follows:
>
> $$
> \begin{aligned}
> \tilde u^\text{reg} _{t _k}(x \mid y) &= \hat u _{t _k}(x \mid y) + \underbrace{w (\hat u _{t _k}(x \mid y) - \hat u _{t _k}(x))} _{\text{CFG boosting term}} - \underbrace{w\gamma \left(s^c _{t _k} \sum^k _{s=0} \beta^s(\hat u _{t _{k-s}}(x \mid y)) - s^u _{t _k} \sum^k _{s=0} \beta^s(\hat u _{t _{k-s}}(x))\right)} _{\text{EMA conditional term}}
> \end{aligned}
> $$
>
> As shown in the equation, REG clearly separates the guidance mechanism into two parts. The CFG boosting term remains identical to the original CFG formulation, preserving both the conditional and quality components without arbitrary down-scaling. The key contribution comes from the EMA conditional term, which leverages the smoothing property of standard EMA ($\beta=0.9$). By subtracting these smoothed historical predictions—which effectively represent the low-frequency conditional component—from the guidance signal, REG cancels out the conditional bias. This structural subtraction allows the method to selectively enhance the quality component, enabling stable sampling even at high guidance scales.

---

> ### Author Response · Authors · 2025-11-23
>
> **C2. Concerns regarding Quantitative Results**
>
> > - **W.2** Some of the ...
> > - **Q.2** Do you have ...
>
> Regarding the IG notation, in EDM2, since $t=\sigma$, we denoted them as $t_{min}, t_{max}$ following the EDM2 convention to align with standard literature. However, we have changed this to the more general $\sigma_{min}, \sigma_{max}$ and updated the parameters to the corresponding sigma values. The reason IG and CFG perform similarly in SDXL is likely that the IG settings in the paper were applied only between 0.28 and 5.42 ("StableDiffusionSigmas" [14.615, 6.475, 3.861, 2.697, 1.886, 1.396, 0.963, 0.652, 0.399, 0.152, 0.0]). Outside this range (only a few steps at the beginning and end), it is identical to CFG, which likely resulted in small differences. We suspect this is why it appears similar to CFG. Additionally, we have updated the notation to sigma for SD3 as well. Regarding FDG, the SDXL hyperparameter was a clerical error carried over from the EDM2 description; the experimental results were correctly entered using the standard setting. We have double-checked the entire table and corrected it for clarity.
>
> Regarding the results previously shown in Table 6 (now presented as Figure 18), it is important to note that REG is explicitly designed to counteract the over-accumulation of conditional components at high guidance levels. Consequently, REG's optimal operating point (peak) naturally shifts to a higher guidance scale compared to CFG. We apologize that our original sweep in Table 6 did not sufficiently cover this higher range to capture REG's true peak. To address this, we have expanded the analysis with finer granularity and extended the range to higher guidance scales, as detailed in the updated table below. These results clearly demonstrate that at its true peak, REG outperforms CFG in both image quality (FID) and diversity (Recall). Furthermore, we provided an additional trend analysis table for DiT-XL, which consistently reaffirms REG's superior diversity-preserving property across different architectures.

---

> ### Author Response · Authors · 2025-11-23
>
> Table: Extended EDM2-S Table
> | CFG | 1.2 | 1.4 | 1.6 | 1.8 | 2 | 2.2 | 2.4 | 2.6 | 2.8 | 3 | 3.2 | 3.4 | 3.6 | 3.8 | 4 | 4.2 |
> | ---------- | ------ | ------ | ------ | ------ | ------ | ------ | ------ | ------ | ------ | ------ | ------ | ------ | ------ | ------ | ------ | ------ |
> | **FID** | 4.04 | 3.45 | 4.82 | 6.89 | 9.14 | 11.34 | 13.38 | 14.98 | 16.53 | 17.73 | 18.78 | 19.73 | 20.47 | 21.15 | 21.70 | 22.15 |
> | **FD_DINOv2** | 143.18 | 102.36 | 81.25 | 72.87 | 70.98 | 72.61 | 76.12 | 80.62 | 85.48 | 90.63 | 95.52 | 100.49 | 105.04 | 109.43 | 113.45 | 117.50 |
> | **Precision** | 0.44 | 0.54 | 0.61 | 0.67 | 0.70 | 0.73 | 0.75 | 0.76 | 0.77 | 0.78 | 0.78 | 0.78 | 0.78 | 0.78 | 0.78 | 0.78 |
> | **Recall** | 0.28 | 0.24 | 0.21 | 0.19 | 0.17 | 0.16 | 0.14 | 0.13 | 0.12 | 0.11 | 0.11 | 0.10 | 0.09 | 0.09 | 0.09 | 0.08 |
>
> | REG | 1.2 | 1.4 | 1.6 | 1.8 | 2 | 2.2 | 2.4 | 2.6 | 2.8 | 3 | 3.2 | 3.4 | 3.6 | 3.8 | 4 | 4.2 |
> | ---------- | ------ | ------ | ------ | ------ | ------ | ------ | ------ | ------ | ------ | ------ | ------ | ------ | ------ | ------ | ------ | ------ |
> | **FID** | 5.17 | 3.70 | 3.41 | 3.87 | 4.80 | 6.02 | 7.37 | 8.74 | 10.03 | 11.26 | 12.47 | 13.52 | 14.48 | 15.48 | 16.43 | 17.12 |
> | **FD_DINOv2** | 166.69 | 131.80 | 108.06 | 91.81 | 81.25 | 75.13 | 72.06 | 70.71 | 70.81 | 72.01 | 73.87 | 76.12 | 78.93 | 81.53 | 84.67 | 87.80 |
> | **Precision** | 0.42 | 0.48 | 0.53 | 0.58 | 0.62 | 0.66 | 0.68 | 0.71 | 0.72 | 0.74 | 0.75 | 0.76 | 0.77 | 0.78 | 0.78 | 0.78 |
> | **Recall** | 0.31 | 0.27 | 0.25 | 0.23 | 0.22 | 0.21 | 0.19 | 0.18 | 0.17 | 0.16 | 0.15 | 0.15 | 0.14 | 0.14 | 0.14 | 0.13 |
>
> DiT-XL peak performance
>
> | | FID | Recall |
> | :--- | --- |  ---: |
> | CFG ($w=1.4$) | 3.45 | 0.24 |
> | REG ($w=1.6$) | **3.41** | **0.25** |

---

> ### Author Response · Authors · 2025-11-23
>
> **C3. Concerns regarding Presentation**
>
> > - **W.3** The images are ...
>
> We sincerely appreciate your constructive feedback. We regret any confusion caused by the presentation issues. We have adjusted the layout to ensure a seamless narrative flow. In the revised manuscript, we have refined the writing throughout and improved the overall presentation to enhance clarity and readability. We have also enhanced the image resolution and quality visualizations as suggested.
>
>
> **C4. EMA Norm calculation analysis through sampling steps**
>
> > - **Q.1** Have you encountered ...
>
> To empirically validate this, we visualized 25 randomly selected samples in Figure 32 instead of the average over the ImageNet 50k dataset, in order to inspect individual trends. As shown in the figure, the norm trajectories remain numerically stable up to the final sampling steps (80 steps) without any divergence.
>
> Technically, as detailed in Appendix C, we incorporate a small constant ϵ=1e-8 into the denominator during the norm-scaling step. This safeguard effectively prevents division by near-zero values, ensuring numerical stability even when the signal magnitude approaches zero toward the end of sampling.

---

### Official Review · Reviewer_CWT2 · 2025-10-31

**Soundness:** 3
**Presentation:** 2
**Contribution:** 2
**Rating:** 4
**Confidence:** 4

**Summary:**

This paper proposes using an **Exponential Moving Average (EMA) of the model's predictions as a substitute for an explicit weak guidance model** in diffusion-based sampling. The premise is that the EMA acts as a low-pass filter, smoothing high-frequency details and thus approximating the output of a "weaker" or less-refined model. The authors claim that this approach, which they term REG, can **improve diversity preservation and sampling quality without the need to train or store additional models**.

**Strengths:**

1. **Novelty of the Approach:** The perspective of **achieving weak guidance without training auxiliary models is compelling**. The original "Weak Guidance" paper highlighted the difficulty of this, and prior strategies for defining a suitable weak model were not entirely convincing.

2. **Clarity:** The paper is **generally well-written and easy to understand**.

**Weaknesses:**

1. **Image Quality:** The **figures in the paper are of poor quality**. They appear to be **heavily compressed and are too small** to properly assess the visual results. For a paper where image quality is a critical component of the evaluation, this is a significant drawback. For example, in Figure 1, it is impossible to discern any meaningful difference between the "no CFG" and "REG" samples. **Higher-resolution images are essential**.

2. **Text-to-Image Metrics:** The authors evaluate T2I generation **using only FID on the COCO dataset**. It is widely known that **FID on COCO is not a reliable metric for T2I**, as improvements in user preference or text alignment do not necessarily correlate with FID improvements. The **evaluation would be much stronger with a more comprehensive set of metrics**, such as GenEval, DPG-Bench, and human-preference-based aesthetic scores (e.g., ImageReward, HPSv2, PickScore) evaluated on standard benchmarks like PartiPrompts or DrawBench.

3. **Limited Model Evaluation:** For Text-to-Image, the method is **only evaluated on SD3**. To substantiate the claim that this method can be broadly used, it would **strengthen the paper to include results on a wider variety of models** (e.g., SD 1.5, SD 2.1, SDXL, PixArt-alpha, SANA, FLUX etc.). Similarly, for ImageNet, evaluating on other model families, like DiT, would be beneficial.

4. **Comparison to Auto-Guidance:** The authors **claim their method can replace auto-guidance, but the results presented seem to contradict this**. The reported FID is **on par with *naive* guidance... and significantly worse than the auto-guidance method** it claims to replace. It is unclear what the advantage of this more complex pipeline is.

5. **Data Presentation:** **Tables 1 and 2 would be much easier to interpret if presented as plots**. For the ImageNet results, an FID vs. IS curve would be a more informative way to show the trade-offs.

6. **Parameter Sensitivity:** The paper would **benefit from an analysis of the sensitivity to the EMA parameters** ($\tau_c$ and $\tau_u$). How do different values affect the final prediction? Furthermore, it's worth exploring whether the same EMA parameter is optimal for all timesteps.

7. **Increased Complexity:** This method **introduces additional complexity at test time by requiring the tuning of four parameters**: $\tau_c$, $\tau_u$, $\omega$, and $\gamma_c$.

8. **Context in Prior Work (Missing Citation):** The paper **misses a key citation**: "Don't drop your samples! Coherence-aware training benefits Conditional diffusion" (Dufour et al., CVPR 2024). This work also presents a single-model weak guidance strategy. ... **A discussion of this prior work is needed**.

**Questions:**

**Formulation of Guidance:** Substituting Equation 6 into Equation 7 yields:

   $$
   \tilde{\epsilon}(x) = \epsilon(x) + \gamma_c (1-\omega) (\epsilon^m(x|y) - \epsilon^m(x)) + \omega(\epsilon(x| y) - \epsilon(x))
   $$

   This **formulation seems counter-intuitive**, as increasing $\omega$ (weight of the full model's guidance) simultaneously *decreases* the weight of the EMA model's guidance. The **dependency between these two guidance vectors may be problematic**. An **exploration of alternative, independent parameterizations could be valuable**. For example:

   * **Option 1:** $\tilde{\epsilon}_t(x) = \epsilon(x) + \omega_1 (\epsilon^m(x|y) - \epsilon^m(x)) + \omega_2(\epsilon(x| y) - \epsilon(x))$

   * **Option 2:** $\tilde{\epsilon}_t(x) = \epsilon_t(x) + \omega_1 (\epsilon^m(x|y) - \epsilon(x)) + \omega_2(\epsilon(x| y) - \epsilon^m(x))$

   Exploring these parameterizations could lead to improvements.

---

> ### Author Response · Authors · 2025-11-23
>
> We sincerely thank you for your time and effort in reviewing our paper. We appreciate your positive comments on:
>
> - the compelling perspective of analysing and defining weak guidance without training auxiliary models
> - the overall clarity and readability of the manuscript
>
> Please find our responses to your comments below.
>
> ---
>
> **C1. Concerns regarding Presentation**
>
> > - **W.1** Image Quality ...
>
> We have replaced all images in the paper with high-quality graphics to address the resolution issues. We apologize for the confusion caused by the initial presentation. Regarding the comment on Line 240, we clarify that the sentence appeared incomplete due to a formatting issue where the text was visually interrupted by Figure 3. We have adjusted the layout to ensure the text flows continuously and clearly. In the revised manuscript, we have refined the writing and improved the overall presentation to enhance clarity and readability.
>
> **C2. Text-to-Image Metrics**
>
> > - **W.2** Text-to-Image Metrics ...
>
> We understand your concern regarding the reliance on FID and COCO. However, datasets like PartiPrompts and DrawBench consist only of prompts without ground truth images, evaluating the calculation of diversity metrics infeasible. Furthermore, metrics such as GenEval, DPG-Bench, ImageReward, HPSv2, and PickScore focus primarily on alignment or human preference, which makes them unsuitable for demonstrating the Quality-Diversity Trade-off. Since our proposed method aims to highlight the Quality-Diversity Trade-off, it was essential to focus on benchmarks where both metrics can be quantitatively assessed. Therefore, in addition to FID, we conducted experiments using Precision and Recall (now reported in Figures 6 and 8 and Table 1) to objectively evaluate diversity, and we kindly ask you to review these results.
>
> **C3. Limited Model Evaluation**
>
> > - **W.3** Limited Model Evaluation ...
>
> We respectfully clarify that the experiments on SDXL were already included in our original submission. However, acknowledging that these results might have been missed during the review process, we have significantly reorganized the layout to enhance their visibility (now explicitly highlighted in Figures 5, 6, 13, 17, and 24 and Tables 4 and 5).
>
> Regarding the ImageNet evaluation, in addition to our existing comprehensive benchmarks on EDM2-s, and EDM2-xxl models, we have conducted additional experiments on DiT-XL as suggested. We evaluated guidance scales ranging from 1.2 to 4.2, covering the official CFG setting of $w=1.5$. We additionally included the evualtion result from the ImageNet. These trends are now reported in Figure 31. As shown in Figure 31, REG consistently improves sample quality (FID, FDDINOv2) while preserving diversity (Recall). Furthermore, REG outperforms CFG in terms of peak performance in both FID and Recall, as summarized in the table below.
>
> | | FID | Recall |
> | :--- | --- |  ---: |
> | CFG ($w=1.4$) | 3.45 | 0.24 |
> | REG ($w=1.6$) | **3.41** | **0.25** |
>
> **C4. Comparison to Auto-Guidance**
>
> > - **W.4** Comparison to Auto-Guidance ...
>
> Regarding the comparison with Auto-Guidance (AG), while the training-free REG method may not explicitly replicate the alignment mechanism of AG (which utilizes a weak conditional prediction for EDM2-s), AG comes with a significant drawback: it nearly doubles GPU memory usage and inference latency by requiring two models simultaneously in Text-to-Image generation. In contrast, our work analyzes the LPF characteristics of EMA to provide an efficient alternative that preserves diversity without such overhead. And we proved that diversity is preserved based on this property (Fig 5, 6, 7 and 8). Also, our approach operates within the standard CFG framework, where conditional and unconditional predictions can be derived from a single model.
>
>
> **C5. Data Presentation**
>
> > - **W.5** Data Presentation ...
>
> Following your recommendation, we have visualized the data from Tables 1 and 2 as plots to improve clarity (now presented as Figures 6 and 8). We appreciate the suggestion as the trends are much clearer. For ImageNet, we have also included the FID-IS curve (now reported in Figure 18) as advised. Although the FID-IS curve for REG shows a similar U-shaped trend to CFG, the REG curve is positioned further to the bottom-right compared to the CFG curve, indicating a better sample quality and diversity trade-off.

---

> ### Author Response · Authors · 2025-11-23
>
> **C6. Concerns regarding the Parameter Sensitivity, Increased Complexity and REG formulation**
>
> > - **W.6** Parameter Sensitivity ...
> > - **W.7** Increased Complexity ...
> > - **Q.1** This formulation seems ...
>
> Regarding the parameters you mentioned ($\tau_c, \tau_u, w, \gamma_c$), we believe you are referring to $\beta_c, \beta_u, w, \gamma$. We acknowledge that defining $\beta_c$ and $\beta_u$ separately may have caused unnecessary complexity, as they effectively utilize the same value. To simplify the formulation, we have unified them into a single parameter $\beta$, fixed at 0.9. Experiments exploring different $\beta_c$ and $\beta_u$ values are provided in Appendix I (now reported in Figures 10 and 26) for reference. Regarding the $\beta$ value over timesteps, we performed a local grid search around 0.9 and found no significant differences. We have added results for various timesteps in the current best setting (now reported in Figure 27). Regarding $\gamma$, we designed it to depend on $w$ (specifically, scaling linearly with $w$) to minimize the search space. In our experiments, a coefficient of approximately from 0.4 to 0.6 was effective for most models. Consequently, $w$ remains the sole parameter requiring tuning, typically set to values higher than standard CFG scales. While Equation 7 may initially appear counter-intuitive, this design is intentional. Our goal is to maintain diversity by canceling out the conditional component using the EMA term, where the impact of this cancellation naturally intensifies with increasing $w$ (the weight of the full model's guidance). Regarding the suggested "Option 1," while we considered this direction, we intentionally coupled $\gamma$ with $w$ to reduce hyperparameter tuning costs. This allowed us to verify our hypothesis by adjusting only the guidance scale without re-tuning $\gamma$. Ultimately, the REG formulation was constructed to align with the difference of conditional and weak conditional structure found in AG.
>
> **C7. Context in Prior Work**
>
> > - **W.8** Context in Prior Work ...
>
> We thank the reviewer for suggesting the paper ``Don't drop your samples!'' (Dufour et al.). While we recognize the relevance of this study, we excluded it from our main benchmarks due to a fundamental difference in scope. As indicated by the title, that paper primarily focuses on training diffusion models without removing weakly annotated datasets. In contrast, our proposed REG is strictly a training-free guidance method designed for diversity preservation at inference time. We are aware that their sampling method shares a similar formulation with CFG and includes a coherence-aware component relevant to diversity. However, their approach prioritizes conditional fidelity and, crucially, necessitates model training/fine-tuning. Consequently, a direct comparison with our training-free method is not methodologically appropriate. Instead, we have incorporated a discussion of this work in the Related Work section (Appendix E), positioning it alongside other relevant methods.

---

### Official Review · Reviewer_3GFE · 2025-11-02

**Soundness:** 3
**Presentation:** 1
**Contribution:** 3
**Rating:** 6
**Confidence:** 4

**Summary:**

The paper observes that the guidance signal from CFG resulted in suppressed diversity, over saturation and layout variation when the guidance scale is too large. This is due to over-amplification of the semantic components leading to over-saturation. The work proposes to solve these problem by applying EMA. However, directly applying EMA on the velocity causing blurry information. The method applies EMA on the guidance term where the guidance components such as conditional term and unconditional term are averaged over the timesteps. This scheme helps to boost semantic shift and avoid amplication on perceptual quality. Hence, reducing over-saturation.

**Strengths:**

1. The method is intuitive
2. The experimental results show some significant improvements.
3. The experiments and analysis seem comprehensive

**Weaknesses:**

1. The paper is written in rush with a lot of grammars and typos. Seriously, in line 240, the section has not completed.
2. The visualization of the whole paper does not look very good.

**Questions:**

Please see the weaknesses

---

> ### Author Response · Authors · 2025-11-23
>
> We sincerely thank you for your time and effort in reviewing our paper. We appreciate your positive comments on:
>
> - the intuitive method utilizing the LPF characteristics of EMA to address the diversity-preserving issue in CFG
> - the significant performance improvements demonstrated in the experimental results
> - the comprehensive experiments and analysis
>
> Please find our responses to your comments below.
>
> ---
>
> **C1. Concerns regarding Presentation**
>
> > - **W.1** The paper is ...
> > - **W.2** The visualization ...
>
> We sincerely appreciate your feedback. We apologize for the confusion caused by the presentation issues.
>
> Regarding the comment on Line 240, we clarify that the sentence appeared incomplete due to a formatting issue where the text was visually interrupted by Figure 3. We have adjusted the layout to ensure the text flows continuously and clearly.
>
> In the revised manuscript, we have refined the writing and improved the overall presentation to enhance clarity and readability. We have also enhanced the quality of the image visualizations as suggested.

---

### Author Response · Authors · 2025-11-23

Dear Area Chair and Reviewers,

We sincerely thank you for reviewing our paper and for your insightful comments and valuable feedback. We appreciate the positive remarks emphasizing the novelty of our work and the advantages of our proposed method:

* **Novelty and Intuitive Nature**: The perspective of utilizing EMA as a low-pass filter to approximate weak guidance is highlighted as novel and interesting (3GFE, CWT2, gP7p, uiNj).
* **Simplicity and Efficiency**: The method is noted for being training-free, simple to implement, and not requiring auxiliary models (CWT2, gP7p).
* **Addressing CFG Limitations**: The approach effectively addresses key limitations of standard CFG, such as over-saturation and the diversity-quality trade-off (3GFE, gP7p).

We have significantly updated the manuscript to address your concerns. Key updates include:

* **Presentation Improvements**: We replaced all figures with **high-resolution graphics**, corrected formatting issues, and improved experimental visualization (e.g., converting tables to plots) for better readability.
* **Expanded Experiments**: We added **DiT-XL** results and **extended the EDM2-s evaluation** (Figures 18 and 31). We also included qualitative comparisons following the AG setting (Figure 30) and FID-IS Curves (Figures 18, 30, and 31) to demonstrate the generalizability of REG and its improvement over CFG.
* **Theoretical Clarifications**: We included a detailed comparison with APG in **Appendix H**.
* **EMA Stability Analysis**: We included an analysis of norm stability across sampling timesteps (**Figure 32**).

We hope these updates have satisfactorily addressed your concerns, and we thank you for your consideration.

Yours sincerely,

Authors

---

### Meta-Review · Area_Chair_DYx5 · 2026-01-09

**Summary:**

This paper introduces Rectified EMA Guidance, a training-free method that mitigates diversity loss and oversaturation in diffusion models by leveraging exponential moving averages during sampling. Evaluated on benchmarks like EDM2 and SDXL, the approach effectively preserves structural richness and improves sample quality without additional training costs.

The paper has received very insightful, constructive and detailed reviews, and the authors have performed hard work to implement the rebuttal. After rebuttal, reviewers actively participated into the discussion phase, and recognized that several concerns were addressed properly. However, based on the reviewers' final comments, there are still critical aspects outstanding, covering weaknesses in complexity, effectiveness and empirical limitations. Therefore, the paper still needs a major revision.

**Reviewer Concerns:**

While reviewers generally appreciate the novelty and theoretical grounding of REG, as well as its ease of implementation, the paper faces significant criticism regarding its presentation. All reviewers highlighted severe writing and formatting issues, including typos and low-quality visualizations. Specifically, Reviewers CWT2 and uiNj questioned the method's sensitivity to hyperparameters and its increased tuning complexity, while also expressing skepticism regarding the empirical comparison with auto-guidance. Additionally, Reviewer CWT2 noted limitations in the evaluation benchmarks, suggesting that more comprehensive metrics and models are needed to substantiate the claims.

**Reviewer Scores:**

Reviewer 3GFE: Only concerns about writing quality which have been addressed.
Reviewer CWT2: Concerns about hyperparameter sensitivity and complexity, comparison with AG, and evaluation benchmarks remain. Not fully addressed.
Reviewer gP7p: Concerns about discussion around APG, sensitivity analysis. Not fully addressed.
Reviewer uiNj: Concerns about hyperparameter sensitivity and complexity, comparison with AG. Not fully addressed.

---

### Decision · Program_Chairs · 2026-01-26

Reject